

# Temporal variations of surface mass balance over the last 5000 years around Dome Fuji, Dronning Maud Land, East Antarctica

Ikumi Oyabu[1], Kenji Kawamura[1,2,3], Shuji Fujita[1,2], Ryo Inoue[2], Hideaki Motoyama[1,2], Kotaro Fukui[4], Motohiro Hirabayashi[1], Yu Hoshina[5], Naoyuki Kurita[6], Fumio Nakazawa[1,2], Hiroshi Ohno[7], Konosuke Sugiura[8], Toshitaka Suzuki[9], Shun Tsutaki[1,2], Ayako Abe-Ouchi[10], Masashi Niwano[11], Frédéric Parrenin[12], Fuyuki Saito[3], Masakazu Yoshimori[10]

[1] National Institute of Polar Research, Tokyo 190-8518, Japan
[2] Department of Polar Science, The Graduate University of Advanced Studies, SOKENDAI, Tokyo 190-8518, Japan
[3] Japan Agency for Marine Science and Technology, Yokosuka 237-0061, Japan
[4] Tateyama Caldera Sabo Museum, Toyama, 930-1405, Japan
[5] The National Museum of Emerging Science and Innovation, Tokyo 135-0064, Japan
[6] Division of Earth and Environmental Science, Graduate School of Environmental Studies, Nagoya University, Nagoya 464-8601, Japan
[7] School of Earth, Energy and Environmental Engineering, Kitami Institute of Technology, Kitami 090-8507, Japan
[8] School of Sustainable Design, University of Toyama, Toyama 930-8555, Japan
[9] Faculty of Science, Yamagata University, Yamagata 990-8560, Japan
[10] Atmosphere and Ocean Research Institute, The University of Tokyo, Kashiwa 277-8564, Japan
[11] Physical Meteorology Research Department, Meteorological Research Institute, Japan Meteorological Agency, Tsukuba, 305-0052, Japan
[12] Université Grenoble Alpes, CNRS, IRD, Grenoble INP, IGE, 38000 Grenoble, France

*Correspondence to*: Ikumi Oyabu (oyabu.ikumi@nipr.ac.jp)

## Abstract

We reconstructed surface mass balance (SMB) around Dome Fuji, Antarctica, over the last 5000 years using the data from 15 shallow ice cores and 7 snow pits. The depth-age relationships for the ice cores were determined by synchronizing them with a layer-counted ice core from West Antarctica (WAIS Divide ice core) using volcanic signals. The reconstructed SMB records for the last 4000 years show spatial patterns that may be affected by their locations relative to the ice divides around Dome Fuji, proximity to the ocean, and wind direction. The SMB records from the individual ice cores and snow pits were stacked to reconstruct the SMB history in the Dome Fuji area. The stacked record exhibits a long-term decreasing trend at -0.037±0.005 kg m$^{-2}$ per century over the last 5000 years in the preindustrial period. The decreasing trend may be the result of long-term surface cooling over East Antarctica and the Southern Ocean, and sea-ice expansion in the water vapor source areas. The multidecadal to centennial variations of the Dome Fuji SMB after detrending the record shows four distinct periods during the last millennium: mostly negative period before 1300 C.E., slightly positive for 1300–1450 C.E., slightly negative for 1450–1850 C.E. with a weak maximum around 1600 C.E., and strong increase after 1850 C.E. These variations are consistent with those of previously reconstructed SMB records in the East Antarctic plateau. The low accumulation rate



periods tend to coincide with the combination of strong volcanic forcings and solar minima for the last 1000 years, but the correspondence is not clear for the older periods, possibly because of the lack of coincidence of volcanic and solar forcings, or the deterioration of the SMB record due to smaller number of stacked cores.

## 1. Introduction

The Antarctic Ice Sheet (AIS) is the largest reservoir of fresh water on the earth and has the potential to increase the global sea level by 58 m (Fretwell et al., 2013). The AIS mass balance is determined by the sum of surface mass balance (SMB), basal melting of ice shelves and peripheral ice discharge into the Southern Ocean, and basal melting at bedrock (e.g., Rignot et al., 2019). The SMB is the sum of surface mass gains (mostly snowfall deposition), surface mass loss (sublimation, water

run-off and evaporation) and blowing snow redistribution (e.g., Lenaerts and Van den Broeke, 2012; Van Wessem et al., 2018; Agosta et al., 2019). Recent satellite remote sensing techniques combined with modeling of regional climate and glacial isostatic adjustment have revealed that the AIS has been losing its mass over the past two decades (e.g., The IMBIE team, 2018; IPCC, 2019, 2021). The ice mass loss is enhanced in the Antarctic Peninsula and West Antarctica mainly due to basal melting of ice shelves and the acceleration of glaciers flow (e.g., The IMBIE team, 2018; Rignot et al., 2019; Schröder

et al., 2019; Velicogna et al., 2020). On the other hand, it is estimated that some parts of Antarctica have been gaining ice mass, possibly because of the increase in snow accumulation. In particular, the ice mass in Dronning Maud Land (DML) significantly increased over the past decade (e.g., Velicogna et al., 2020). Although in-situ observations (e.g., snow stake measurements), remote sensing observation (e.g., Velicogna et al., 2020) and regional climate models (e.g., Mottram et al., 2021) are useful for evaluating the spatiotemporal changes in AIS MB and SMB, they are available only over the last few

decades. However, the AIS also changes in much longer time scales than those covered by the direct observations (e.g., Colleoni et al., 2018), the long-term records of the SMB are needed to better understand the AIS mass changes (Stokes et al., 2022). In this study, we use the term accumulation or accumulation rate to refer to positive SMB, because the long-term SMB is generally positive on the interior part of AIS.

Firn and ice cores have provided the AIS SMB histories over the last few centuries to millennia. Thomas et al. (2017) compiled 79 records of Antarctic ice-core-based SMB records over the past 1000 years. The analysis of the total AIS SMB derived from the database suggested that snow accumulation had significantly increased in the 20th century (over the past ~50 and ~100 years) (Thomas et al., 2017). A newer SMB reconstruction over the last 200 years by combining the above dataset and reanalysis data suggests accelerating the increase in AIS accumulation rate during the latter half of the 20th

century (Medley and Thomas, 2019). In the East Antarctic Plateau (EAP), the ice-core-based SMB reconstructions do not show a uniform trend. At the Kohnen station (EPICA DML), the increasing trend of accumulation rate was reported over the past 200 years (Oerter et al., 2000; Hofstede et al., 2004; Altnau et al., 2015). Along the Japanese-Swedish traverse route between the Wasa and Dome Fuji stations, Fujita et al. (2011) combined shallow cores and ice-radar isochrons, and found a





strong relationship between the SMB and local surface topography, and a higher accumulation rate in the latter half of the
20th century than the preindustrial Holocene average by 15%. A significant increase in accumulation rate in the 20th century
was also found from a firn core at the Dome Fuji station (Igarashi et al., 2011). At the Dome C station, stake measurements
and firn cores showed that the accumulation rate over the last 200 years increased by ~30% with respect to the average over
the last 5000 years, based on the reconstruction using stable water isotope record as a proxy for SMB (Frezzotti et al., 2005).
At Vostok, the accumulation rate after 1950 C.E. is larger than the average between 1260 and 1600 C.E. (Osipov et al.,
2014). On the other hand, along the Norwegian-USA traverse routes between the coastal DML and the South Pole, Anschütz
et al. (2009, 2011) found no consistent trends in SMB since 1963 C.E.; some sites showed an increase in accumulation rate
while others showed a decrease. The same study found decreasing trends in accumulation rate for the sites above 3200 m of
elevation. The apparently inconsistent trends of the SMB over the EAP may be related to their locations relative to ice ridges
and prevailing wind directions (Fujita et al., 2011) or the influences of atmospheric patterns such as Southern Annular Mode
(Medley and Thomas, 2019).

On the multi-centennial time scale, a negative trend in SMB since 1000 C.E. was found in the SMB composite of four
regions with long records: West Antarctic Ice Sheet (WAIS) and coastal sites in Wilkes Land, Weddell Sea coast, Victoria
Land (Thomas et al., 2017). However, only four ice core records actually extend to the 1000 years (WAIS Divide, Law
Dome, Berkner Island and Roosevelt Island cores), and the vast interior of the East Antarctic Ice Sheet (EAIS) is not
represented in the 1000-yr composite (Thomas et al., 2017). The authors suggested that small changes in the accumulation
rate in the EAP, although it is relatively low, could change the sign and significance of the total Antarctic SMB trend,
because the total area of the EAP accounts for about half of that of the AIS. Thus, reliable long-term SMB records from EAP
are particularly valuable.

There are several reasons for the lack of reliable, continuous and long-term SMB reconstructions from EAP. (1) The top part
(surface to several meters) of shallow cores are extremely fragile, which makes it challenging to precisely measure the
density and assign the depths of age markers for the last few hundred years. (2) Snow deposition is not spatially and
temporally uniform, and annual layers can be eroded (Kameda et al., 2008), thus an SMB reconstruction from one core may
have large uncertainty. (3) Density profiles in the low accumulation area are highly variable, especially near the surface
(Weinhart et al., 2020), introducing large uncertainty in the estimated masses. (4) Sparseness of volcanic eruption records
before 1000 C.E. and high-resolution ice-core datasets, combined with occasional lack of annual layers in the low-
accumulation cores, make it difficult to precisely and densely date the ice cores. Thus, the existing SMB reconstructions in
EAP over more than several thousand years are based on a proxy method (from stable water isotopes; e.g., Parrenin et al.,
2016), or on a small number of isochrons from shallow ice radar measurements (e.g., Fujita et al., 2011; Cavitte et al., 2018,
2022).



In this study, we reconstruct the SMB histories around Dome Fuji over the last 5000 years using data from 7 snow pits and 15 shallow ice cores retrieved by successive Japanese Antarctic Research Expedition (JARE). We determined the depth-age

relationships of the ice cores by synchronizing them with the WAIS Divide ice core, whose chronology is accurately determined by annual layer counting, using common volcanic signals (Sigl et al., 2014, 2016). We discuss millennial-scale trends and centennial-scale variabilities of SMB in relation to the climate variabilities during the mid- to-late-Holocene.

## 2. Materials and methods

### 2.1 Study area and samples

The study area of this study is the inland Dronning Maud Land around the Dome Fuji station (77.316 S, 39.701 E, 3810 m a.s.l.) with a radius of about 300 km (Fig. 1). Dome Fuji is the second highest summit of the AIS and is located at the junction of four ice divides (Fig. 1a). Spatial and temporal variability of SMB around the ice divide near Dome Fuji is characterized as follows. The large-scale spatial pattern of the SMB depends on the surface elevation, continentality, and large-scale surface topography (e.g., positions of ice divides) in combination with a dominant trajectory of airmass with

moisture. Around Dome Fuji, SMB is larger on the windward (mostly north-eastern) side of ice divides than on the leeward side (Suzuki et al., 2008; Fujita et al., 2011; Tsutaki et al., 2022). In addition, local variations in surface topography, which may be influenced by the bedrock topography and ice flow (Fujita et al. 2011; Tsutaki et al., 2022), also affect the precipitation patterns and snow redistribution, thus the spatial patterns of SMB (Furukawa et al., 1996, Van Liefferinge et al., 2021).


In this study, we use 7 snow pit samples and 15 ice cores collected between 1993 and 2019 to reconstruct accumulation rates. The name and locations of each site are shown in Figure 1 and Table 1. The locations of the sites are mostly distributed within 50 km from Dome Fuji (Fig. 1b), with additional sites further away (200–400 km), both on the windward and leeward sides of the main ice divides with respect to the origin and transport paths of moisture. At the Dome Fuji station, two deep

ice cores have also been drilled in the 1990s and 2000s to the depths of 2503 m (340 kyr) (DF1) and 3035 m (720 kyr) (DF2), respectively (Watanabe et al., 1997a, 1997b; Dome Fuji Ice Core Project Members, 2017; Motoyama et al., 2021). Two shallow cores (DF1993 and DF2001) are the shallow parts of the DF1 and DF2 deep ice cores. Because the NDF2018 core is the longest and oldest shallow ice core in this study (151 m long, covering ~4800 years), we use the records of DF1 and DF2 deep cores to cover the same period. Four ice cores (DF1, DF2, NDF2018, and NDFN cores) cover more than 4000 years,

three ice cores (DF1997, DFS2010 and DFS2011 cores) cover 1500 years, and six ice cores (MD364, NDF2013, S79, S80, DFNW and DFSE cores) cover 800 years or less. The DF1997 core has poor ice core recovery and uncertain depth assignments for 0–20 m, thus we use the data only below 20 m. The DF1999 core was measured for permittivity (for density estimation), but not for DEP for volcanic matching, thus only used for estimating the average density profile at Dome Fuji.





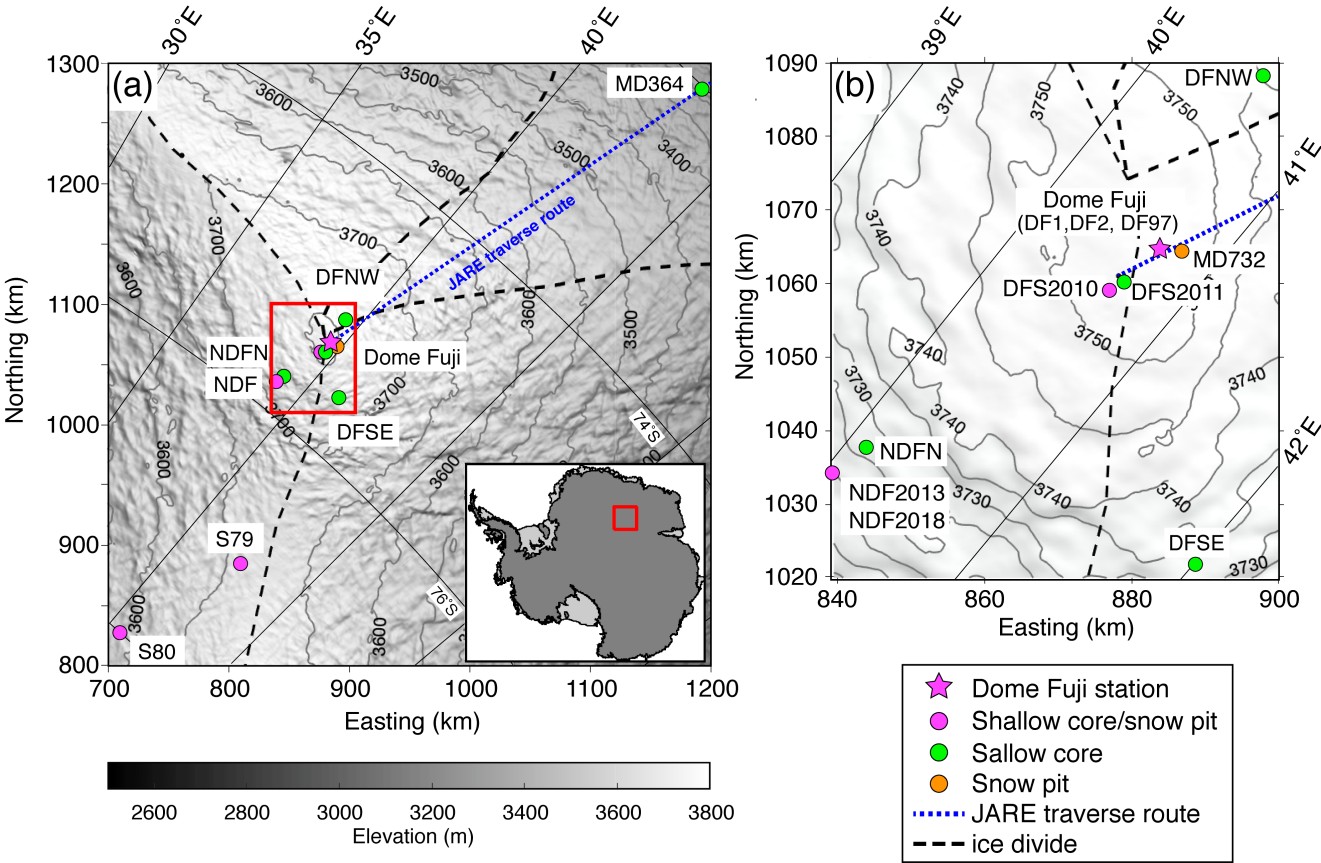

Figure 1: **Locations of ice cores and snow pits (polar stereographic projection). (a) Overall view, and (b) enlarged view. Makers in magenta are the sites with both ice core and snow pit, markers in green are ice core sites, and markers in orange are snow pit sites. Surface elevation contours intervals are (a) 50 m and (b) 5 m (BedMachine, Morlighem et al., 2020). Figures were drawn using an open-source MATLAB toolbox (Antarctic Mapping Tools, Greene et al., 2017).**

Table 1: **Location, elevation, depth and observation date of the shallow cores and snow pits.**

| Site name | Latitude °S | Longitude °E | Elevation (m) | | Depth (m) | Bottom year* (C.E.) | Observation date (yyyy/mm/dd) | |
|---|---|---|---|---|---|---|---|---|
| MD364 | 74.007 | 42.997 | 3353 | core | 80.08 | 400 | 2001/12/25–29 | JARE42 |
| DFNW | 77.071 | 39.531 | 3788 | core | 43.26 | 1200 | 2018/1/5–7 | JARE59 |
| Dome Fuji (DF1993) | 77.316 | 39.701 | 3810 | core | 112.00 | -900 | 1993/12/5–16 | JARE34 |
| Dome Fuji (DF1996) | 77.316 | 39.701 | 3810 | core | ~40 | 1200 | 1996/12/25 | JARE37 |
| Dome Fuji (DF1997) | 77.316 | 39.701 | 3810 | core | 131.00 | -1500 | 1997/10/14–11/14 | JARE38 |
| Dome Fuji (DF1999) | 77.316 | 39.701 | 3810 | core | 108.47 | -800 | 1999/12/12–24 | JARE39 |
| Dome Fuji (DF2001) | 77.316 | 39.701 | 3810 | core | 122.40 | -1200 | 2001/11/19–25 | JARE42 |
| DFS2011 | 77.373 | 39.657 | 3799 | core | 113.00 | -900 | 2011/1/20–25 | JARE52 |
| DFS2010 | 77.395 | 39.617 | 3798 | core | 120.00 | -1000 | 2010/1/15–20 | JARE51 |



| | | | | | | | | |
|---|---|---|---|---|---|---|---|---|
| DFSE | 77.584 | 41.024 | 3779 | core | 41.00 | 1200 | 2017/12/31–2018/1/2 | JARE59 |
| NDFN | 77.736 | 39.118 | 3772 | core | 142.12 | -2400 | 2018/12/14–29 | JARE60 |
| NDF2013 | 77.787 | 39.059 | 3754 | core | 30.80 | 1300 | 2012/12/ 24–25 | JARE54 |
| NDF2018 | 77.788 | 39.054 | 3754 | core | 151.88 | -2900 | 2017/12/19–27 | JARE59 |
| S79 | 79.001 | 42.497 | 3700 | core | 56.88 | 600 | 1997/12/12–15 | JARE38 |
| S80 | 80.000 | 40.501 | 3622 | core | 30.17 | 1400 | 2013/1/2–3 | JARE54 |
| MD364 | 74.007 | 42.997 | 3353 | pit | 1.05 | | 2007/11/30 | JARE49 |
| MD732 | 77.298 | 39.786 | 3785 | pit | 4.02 | | 2007/12/10–11 | JARE49 |
| Dome Fuji | 77.316 | 39.701 | 3810 | pit | ~1.2–3.80 | | 1997/1/18, 2/22, 3/5, 4/4, 5/5, 8/5, 9/10, 10/4, 11/18, 12/26, 2003/2/4 | JARE38, 44 |
| DFS2010 | 77.395 | 39.617 | 3798 | pit | 2.30 | | 2010/1/22 | JARE51 |
| NDF2013 | 77.787 | 39.059 | 3754 | pit | 2.18 | | 2012/12/22 | JARE54 |
| NDF2018 | 77.788 | 39.054 | 3754 | pit | 4.02 | | 2017/12/26–28 | JARE59 |
| S79 | 79.001 | 42.497 | 3700 | pit | 2.06 | | 2013/1/5 | JARE54 |
| S80 | 80.000 | 40.501 | 3622 | pit | 2.36 | | 2012/12/30 | JARE54 |

*Round to the nearest hundred year.

## 2.2 Density

We obtained density profiles of shallow cores by combining published density profiles, bulk densities measured at the field,
and permittivity measured in a laboratory. We briefly describe each method.

### 2.2.1 Bulk density measurements

We use the bulk density data for estimating the SMB for the MD364, S79, DFS2011, NDFN, DFSE and DFNW cores. Bulk density of an ice core piece was calculated from its diameter, length and mass measured at the drilling sites, assuming that it
has a cylindrical shape. The diameter was measured with a caliper at one to three positions, and the mass was measured with an electronic balance. The density of near-surface snow was obtained by excavating a snow pit and sampling the snow along the pit wall with a stainless-steel snow density sampler, which was measured with an electronic balance. The typical dimensions of the density sampler are 60 mm (W) × 56 mm (D) × 30 mm (H) (~100 cm$^3$).

To assess the uncertainty of the in-situ ice core measurements, the bulk density of the NDFN core was also measured between 21 and 64 m (correspond to ~500–700 kg m$^{-3}$) in a cold laboratory at the National Institute of Polar Research (NIPR). The difference between the in-situ and laboratory data likely originates in the harsh measurement environment in the field (wind, lack of time, uneven worktable, etc.), and was 6±15 kg m$^{-3}$ (mean and standard deviation). Because it is difficult to evaluate the errors of the bulk density for different cores, we employ ±15 kg m$^{-3}$ from the NDFN core as the uncertainty of
the in-situ bulk density data for all cores. In addition to the random error, underestimation is common for ice cores from the



Antarctic inland for depths shallower than ~15 m because of the extreme fragility of the cores. For example, the bulk density of the NDF core at ~4 m is underestimated by ~20 % with respect to the density measured from snow pit sampling at the same site. The uncertainty of the snow-pit density measurements performed with similar devices and procedures has been reported as ±4% (Conger and McClung, 2009).


### 2.2.2 Relative permittivity at millimeter wave frequencies

We measured the high-frequency-limit relative permittivity of firn cores using open resonators operating under frequencies from ca. 15 GHz to ca. 40 GHz. Hereinafter, we simply use terms as relative permittivity, permittivity, or $\varepsilon$. $\varepsilon$ were measured at NIPR and converted to firn densities $\rho$ (kg m$^{-3}$) using empirical relations between $\varepsilon$ and $\rho$ (Kovacs et al., 1995; Fujita et al.,

2014). The detailed method for the measurement of $\varepsilon$ is described elsewhere (Fujita et al., 2009, 2014, 2016; Saruya et al., 2022). Briefly, a core sample is cut into slab shape, with a typical thickness of 5–80 mm and a width of 45–65 mm. The sample is scanned with a Gaussian beam of electromagnetic waves with a 1-$\sigma$ diameter of 15–38 mm using an open resonator. With the method, we can detect tensorial values of $\varepsilon$, that is, $\varepsilon$ with electrical field along the vertical of the cores when core ice was in the ice sheet ($\varepsilon_v$), and $\varepsilon$ with electrical field parallel to the horizon ($\varepsilon_h$). The measured $\varepsilon_h$ was converted

to $\rho$ using empirical relations under the temperature of measurements, either -16 (±1.5) °C or -30 (±1.5) °C.

For measurements under -16 °C, we used

$\rho = -5.556\varepsilon_h^3 - 4.0922\varepsilon_h^2 + 494.14\varepsilon_h - 436.121$     (1)     (Fujita et al., 2014).

This relation was applied for cores from 7 sites; the DF1993, DF1996, DF1999, DFS2010, NDF2013 and S80. For the

DF1993, DF1999 and DFS2010 cores, we used published records of $\varepsilon$ (Fujita et al., 2016). For the NDF2018, DFNW, DFSE and NDFN cores, which were measured in 2018–2021, measurements were under -30 °C. We use:

$\rho = -20.15\varepsilon_h^3 + 99.801\varepsilon_h^2 + 243.02\varepsilon_h - 220.57$     (2)     (this study).

The analytical uncertainty of $\varepsilon_h$ is ±0.005, and the overall uncertainty for the density is estimated to be ~ 6–14 kg m$^{-3}$ (Appendix A).


### 2.2.3 Fitting and interpolation

For S79, MD364 and DFS2011 cores, we fitted a polynomial function to the ice-core data below 15 m with a fixed surface density from the pit data due to the underestimation of bulk density at shallow depths. We employed a second-order polynomial function for the S79 core and a fifth-order polynomial function for the MD364 and DFS2011 cores. The degree

of the polynomial functions was chosen to minimize the residual (difference between the data and fitting curve). A second-order polynomial function was sufficient for the relatively short core (S79 core, 56 m), while fifth-order polynomial



functions were necessary for the MD364 (80 m) and DFS2011 (113 m) cores. The surface densities (intercept for the fitting

curves) for the MD364 and S79 sites were determined by averaging the pit densities over the top 0.5 m at the respective sites.

Because there is no pit data for the DFS2011 site, we employ the surface density at DFS2010, which is the nearest

observation site to DFS2011 (2.6 km south of DFS2011). The coefficients for the polynomial functions are described in

Figure 2.

For the DFNW, DFSE and NDFN cores, the permittivities were measured over the top 20 m, thus we used them for the top

20 m without smoothing. For the deeper depths, polynomial functions were fitted to the bulk density data over the whole

cores, and simply used the fitted curve below 20 m. We employ a second-order polynomial function for the DFNW (43 m)

and DFSE (41 m) cores and a fifth-order polynomial function for the NDFN core (142 m). The coefficients for the

polynomial functions are described in Figure 2.

For DF1, DF1997 and DF2 cores, which were drilled at the Dome Fuji station, we constructed a common density profile as

follows. We note that available density data from the DF cores (DF1993, 1996, 1997, 1999 and 2001) were similar to each

other (Fig. 2). Continuous permittivity data was available for the top 3.47 m on the DF1993 core, thus we converted it to

density and used it without smoothing. For the deeper depths, we fitted polynomial functions to discontinuous permittivity-

based density from the DF1993 and DF1999 cores (3.5–112 m) and bulk density data from the DF2001 (10.2–122.2 m) and

DF1 (155.1, 180.6, 230.1, 276.3, 316.9, 362.8 and 400.7 m, T. Kameda, personal communication) cores. The maximum

length of the DF cores (175 m) was too long to fit adequately with a single polynomial function; thus, we divided the depth

range into two sections. The upper section (0–122 m) and the lower section (74.6–400.7 m) were fitted with respective fifth-

order polynomial functions, and the curves were switched at 108.5 m, where they connect smoothly. The resulting functions

are:

$$\rho = 323 + 12.417d - 0.22226d^2 + 0.0029985d^3 - 2.03e\text{-}5d^4 + 5.10e\text{-}8d^5 \text{ (3.47–108.5 m)} \tag{3}$$

and

$$\rho = 195.67 + 12.078d - 0.080521d^2 + 0.0002636d^3 - 4.21e\text{-}7d^4 + 2.62e\text{-}10d^5 \text{ (108.5–175 m)} \tag{4},$$

where $d$ is depth. The uncertainty of the DF density profile thus constructed was estimated as the standard deviation of the

difference between the DF1993 bulk density data and the fitted curves.

For NDF2013 and S80 cores, we combined the pit densities (2.18 m and 2.36 m for NDF2013 and S80, respectively) and the

densities converted from the permittivity.

Figure 2l compares the density profiles from 8 sites (MD364, DFNW, Dome Fuji, DFS2010, DFSE, NDFN, NDF2018 and

S80). The density at a given depth at MD364 is higher than those at the other sites, as expected from the markedly higher

accumulation rate (Herron and Langway, 1980). The density profiles of the sites near the Dome Fuji station (Dome Fuji,

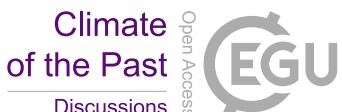

DFS2010, NDFN and NDF2018) are similar to each other, but a closer inspection shows that the densities at NDFN and NDF2018 are consistently higher than those at Dome Fuji and DFS2010 below ~50 m, and the depths to reach 830 kg m⁻³, which is considered as a typical close-off density (Cuffey and Paterson, 2010), are shallower at the former sites than those at the latter sites by ~5 m.


**Figure 2: Density profiles of the studied sites. The black lines are the data used for the calculation of the accumulation rate, with 1σ uncertainty as a grey shade. Colored markers indicate bulk density data. K0, K1, K2, K3, K4 and K5 are coefficients for the quadratic or quintic fitting curves. See main text for coefficients for the fitting curve for the Dome Fuji (eq. 3 and eq.4).**




**Table 2: Surface density from snow pits**

| Site name | Date (yyyy/mm/dd) | Density | | | | | | | |
| --- | --- | --- | --- | --- | --- | --- | --- | --- | --- |
| | | 0–0.5 m | | 0–1.0 m | | 0–1.5 m | | 0–2.0 m | |
| | | Ave (kg m$^{-3}$) | 1 σ | Ave (kg m$^{-3}$) | 1 σ | Ave (kg m$^{-3}$) | 1 σ | Ave (kg m$^{-3}$) | 1 σ |
| Dome Fuji | 1997/1/18 | 322.0 | 23.3 | 310.6 | 26.4 | 319.3 | 30.1 | 322.6 | 28.7 |
| Dome Fuji | 1997/2/22 | 318.8 | 16.5 | 320.7 | 22.1 | | | | |
| Dome Fuji | 1997/3/5 | 311.7 | 16.6 | 317.3 | 18.4 | | | | |
| Dome Fuji | 1997/4/4 | 302.3 | 21.5 | 321.0 | 37.1 | 335.2 | 47.2 | 331.9 | 42.8 |
| Dome Fuji | 1997/5/5 | 325.6 | 27.2 | 322.3 | 25.1 | | | | |
| Dome Fuji | 1997/8/5 | 315.2 | 30.7 | 321.1 | 25.2 | | | | |
| Dome Fuji | 1997/9/10 | 340.7 | 34.5 | 344.4 | 44.8 | | | | |
| Dome Fuji | 1997/10/4 | 335.8 | 30.0 | 361.7 | 52.2 | | | | |
| Dome Fuji | 1997/11/18 | 328.2 | 22.2 | 345.4 | 38.7 | | | | |
| Dome Fuji | 1997/12/26 | 327.9 | 29.3 | 346.9 | 49.6 | 347.1 | 47.2 | 358.2 | 50.1 |
| Dome Fuji | 2003/2/4 | 368.2 | 32.1 | 378.5 | 29.5 | 371.4 | 29.1 | 381.6 | 34.2 |
| MD364 | 2007/11/30 | 416.6 | 50.6 | 424.0 | 50.6 | | | | |
| MD732[a] | 2007/12/10–11 | 316.5 | 22.8 | 330.8 | 28.2 | 338.1 | 29.4 | 343.1 | 29.2 |
| DFS2010 | 2010/1/22 | 313.8 | 21.9 | 313.2 | 23.0 | 316.4 | 21.7 | 319.7 | 22.9 |
| NDF2013 | 2012/12/22 | 349.3 | 37.6 | 343.6 | 33.7 | 346.8 | 36.2 | 345.7 | 32.9 |
| NDF2018 | 2017/12/27–29 | 372.0 | 39.5 | 378.4 | 36.9 | 369.3 | 35.7 | 364.3 | 34.1 |
| NDFN[b] | 2018/12/16–27 | | | 357.5 | | | | | |
| S79 | 2013/1/5 | 331.9 | 42.3 | 335.5 | 34.5 | 336.2 | 34.4 | 335.6 | 30.8 |
| S80 | 2012/12/30 | 335.1 | 27.4 | 336.2 | 28.2 | 342.8 | 26.5 | 340.5 | 27.7 |

[a] Hoshina et al. (2014), [b] Van Liefferinge et al. (2021)

## 2.3 Ice core and snow sample measurements for dating

Annual layer counting is not possible for our samples because some annual layers can be eroded due to the low accumulation
rate (Kameda et al., 2008). Therefore, we used volcanic signals recorded as near-surface conductance of the direct current
(ECM) (Hammer, 1980; Wolff, 2000), high-frequency-limit electrical conductivity (DEP) and non-sea-salt sulfate (nssSO$_4^{2-}$),
or tritium peaks from the past bomb tests to identify the age horizons. We used published data of the ECM of the DF1993



and DF1 cores (Fujita et al., 2015), $nssSO_4^{2-}$ concentrations of the DF2001 core (Motizuki et al., 2014), $nssSO_4^{2-}$ concentrations and tritium contents of the MD732 snow pits (Hoshina et al., 2014), respectively. For other ice cores and

snow pit samples, we newly generated the data with the following methods.

### 2.3.1 DEP

Dielectric profiling, DEP, is a method to measure high-frequency (typically 250 kHz or higher frequency) conductivity to quickly locate positions of volcanic events as acidity spikes in ice cores. The method is given in the literature (Moore and

Paren, 1987; Wilhelms et al., 1998). At NIPR, DEP measurements were performed for DF2, DF1997 , DFS2010, DFS2011, NDF2018, DFNW, DFSE and NDFN cores. The ice cores were measured every 20 mm. The uncertainties for the depth assignment of the DEP measurement is ±0.02 m.

### 2.3.2 Ion concentrations

Ion concentrations were measured at NIPR for the MD364, DF2001, NDF2013, S79 and S80 cores, and snow pit samples from Dome Fuji, DFS2010, NDF2013, NDF2018, S79 and S80. The ice core samples were decontaminated by shaving off their surfaces by ~10% by weight with a pre-cleaned ceramic knife in a clean bench, and placed in particle-free plastic bags. The plastic bags were brought to the clean laboratory at room temperature, and the samples were melted in the bags and analyzed with established procedures (e.g., Goto-Azuma et al., 2019) for the concentrations of $Cl^-$, $SO_4^{2-}$, $NO_3^-$, $F^-$, $CH_3SO_3^-$,

$Na^+$, $NH_4^+$, $K^+$, $Mg^{2+}$ and $Ca^{2+}$ using ion chromatograph (DX-500 or ICS-5000+, Thermo Fisher Scientific). The snow samples, originally collected in clean plastic bottles, were melted at room temperature in the clean room and analyzed for ion concentrations.

We corrected $SO_4^{2-}$ concentration for the sea salt contribution with a common method by assuming that the source of $Na^+$ is

only seawater and sea-salt $SO_4^{2-}$ is always accompanied by $Na^+$:

$$[nssSO_4^{2-}] = [SO_4^{2-}] - 0.25 \cdot [Na^+] \qquad (5)$$

where 0.25 is the ratio of $SO_4^{2-}$ to $Na^+$ in the seawater (Millero et al., 2008).

### 2.3.3 Tritium contents

Tritium contents of the Dome Fuji and NDF2018 snow pit samples were measured at NIPR and Nagoya University, respectively, with the liquid scintillation method (Kamiyama et al., 1989). The snow samples were distilled, mixed with liquid scintillation cocktail and homogenized. The tritium radioactivity was measured with a low background liquid scintillation counter (LSC-LB3, Hitachi at NIPR, or Quantulus 1220, PerkinElmer at Nagoya University). The minimum



detection level is around 7 TU, and the relative uncertainty is less than 5%. Measured values were corrected for radioactive
decay and converted into the data as of the time of sampling (February 2003 for Dome Fuji and January 2018 for NDF2018).

**2.4 Age determination of volcanic layers in the ice cores and snow pits**

The depth of a volcanic layer is defined by the maximum values of measured data (DEP, ECM or nssSO$_4^{2-}$) (Hofstede et al.,
2004). The volcanic signals from low-latitude eruptions in Antarctic snow lag the actual eruption by 1 to 2 years due to the
long-distance transport (Hammer et al., 1980; Cole-Dai and Mosley-Thompson, 1999; Cole-Dai et al., 2000, 2009; Gao et al.,
2006). This depositional lag is not constant and may generate additional uncertainty for the dating. To avoid the uncertainty
of depositional lag, we take advantage of published sulfur data of the WAIS Divide ice core (WDC) with accurate layer-
counting chronology (Sigl et al., 2014, 2016), to which we can precisely synchronize the volcanic signals in our shallow
cores and pit samples. We assumed no depositional lag between WAIS Divide and Dome Fuji. Based on the synchronization
of DF1993 and DF1 cores to WDC by Oyabu et al. (2022a, in review), we identified the eruptions in the other shallow cores
by inter-comparisons of their volcanic signals. We only employed the peaks found in two or more of our ice cores. In total,
we identified 62 volcanic layers for the dating between 1992 C.E. and 3151 B.C.E (Fig. 3). We also confirmed the
consistency of our dating with previous studies that used the same volcanic layers (back to 1260 C.E. for the DF2001 core by
Igarashi et al., 2011; back to ~100 C.E. for the DFS2010 core by Motizuki et al., 2014). We found several common peaks in
our ice cores that are not recorded in the WDC, which were not used for the dating because their depositional age could not
be accurately known. We note that a common 1-cm-thick tephra layer was found in the DF2, NDF and NDFN cores at
133.51, 118.66 and 122.18 m, corresponding to 1613±5, 1613±5 and 1611±4 B.C.E., respectively, according to their
chronologies (section 2.5). The tephra layer is not found in WDC but it is probably the same as those found in the Dome C,
Vostok and South Pole at 132.6, 103.14 and 303.44 m, respectively (Palais et al., 1987; Narcisi et al., 2005), dated around
1550–1650 B.C.E (on Vostok GT4 and EDC2 chronologies, Narcisi et al., 2005). While the synchronization using tephra
layers is generally difficult because of their sparseness and large signals from local eruptions, this tephra layer is easily
visible in the field, and the age of the layer determined here (1613 B.C.E or 3563 yr BP) may be useful as an age constraint
for other shallow ice cores from EAP.

The uncertainty of duration between two neighboring volcanic tie points, which is required for the error estimate of SMB,
consists of the following components: (1) depth of ECM or DEP measurement (±0.01 m as 1σ for each volcanic layer,
corresponding to 0.15–0.35 years), (2) identification of peak positions (±0.01 m as 1σ for each volcanic layer, corresponding
to 0.15–0.35 years), and (3) duration between the two horizons originating from the age uncertainty of WD2014 (<0.75 years
as 1σ) (Sigl et al., 2016). The overall uncertainties for the shallow cores are smaller than 1 year (1σ).






Figure 3: DEP, ECM or nssSO₄²⁻ for our ice cores and sulfur concentration of the WAIS Divide core (Cole-Dai, 2014a, b). The previously published data are the nssSO₄²⁻ data for 7.7–85.49 m (Motizuki et al., 2014), ECM data for the DF1993 and DF1 cores (Fujita et al., 2015), and DEP data for the DF2001 and DF2 cores (Fujita et al., 2015). Vertical bars indicate volcanic tie points.

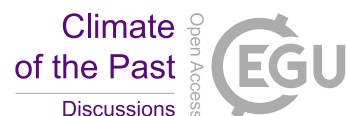


For the snow pit data, we used the sulfate peaks for the Pinatubo eruption (deposition in 1992) for all snow pits except for
Dome Fuji, and sulfate peaks for the Agung eruption (deposition in 1964) and Tritium peaks (deposition in 1966, Fourré et
al., 2006) for MD732, Dome Fuji and NDF2018 snow pits (Fig. 4). It is difficult to determine the depth of the Pinatubo
eruption in the Dome Fuji pit data because there are three possible nssSO$_4^{2-}$ peaks for the eruption within ~0.5 m interval
(around 1.0 m). The error of the duration between the tie points is estimated to be 0.5 yr, and that for the peak depth is
estimated to be 0.1 m (corresponding to ~1 yr), thus the overall uncertainty is ~1.1 yr (1σ).

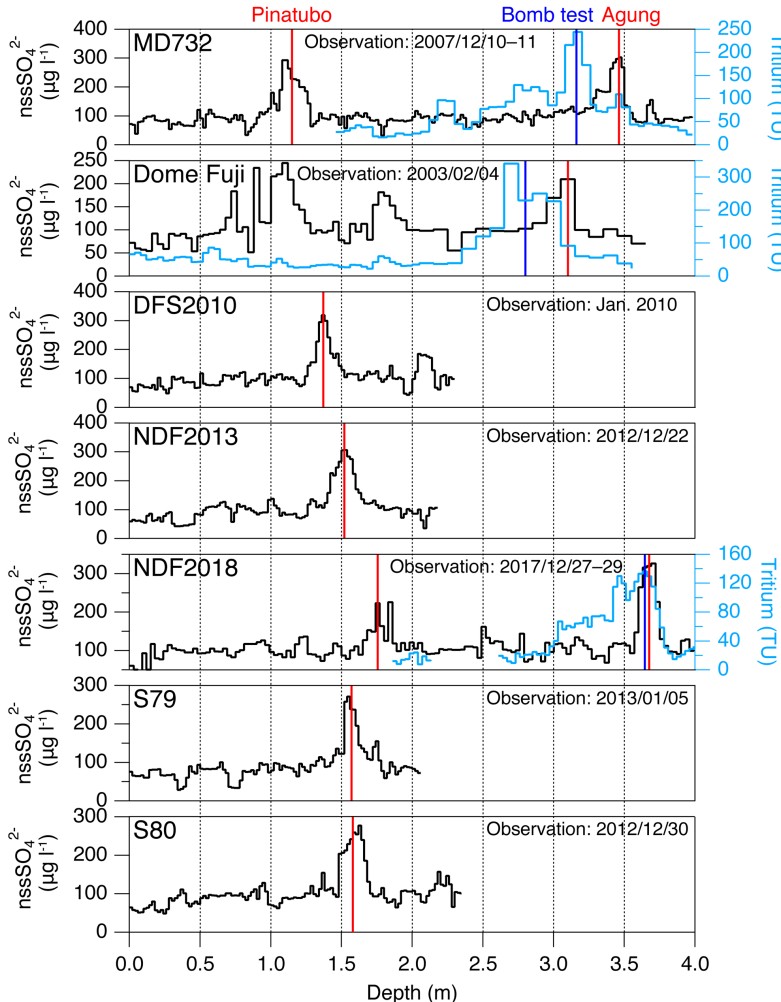


**Figure 4: nssSO$_4^{2-}$ concentration and tritium content for snow pits. The data for MD732 are from Hoshina et al. (2014). Vertical
bars represent the depths of volcanic eruptions (red) and the past bomb test (blue) for age controls.**





## 2.5 Calculation of accumulation rate and its uncertainty

Mean accumulation rates between the dated horizons are calculated by dividing the depth difference between the age markers, corrected for firn density and layer thinning (due to ice flow), by the time span. For the thinning function, we used a published model result for the Dome Fuji core (Parrenin et al., 2007), and applied it to other cores using normalized ice-equivalent depth between the surface and bottom of the ice sheet (Fig. 5). We assumed that the uncertainty of the thinning function linearly increases with depth at a rate of 0.01% m$^{-1}$ (1σ), giving reasonable uncertainty including the assumption of

linear scaling of the Dome Fuji thinning function to other sites (note that the difference between the thinning function between DF and NDF is about 1.5 % at 150 m). The effect of the thinning correction on the resulting accumulation rate is smaller than the overall uncertainty (see below) for the depths shallower than 40 m, roughly corresponding to the last millennium, while it is highly important for the deepest part (4–5 kyr BP) of the DF, NDF and NDFN cores (about three times the overall uncertainty) to discuss the long-term variations of SMB. The overall uncertainty of SMB for a given core

and interval was estimated by using a Monte Carlo approach, in which the density profile, age of horizons and thinning functions were randomly varied 1000 times according to the respective error estimates to calculate the probability distribution of accumulation rate.

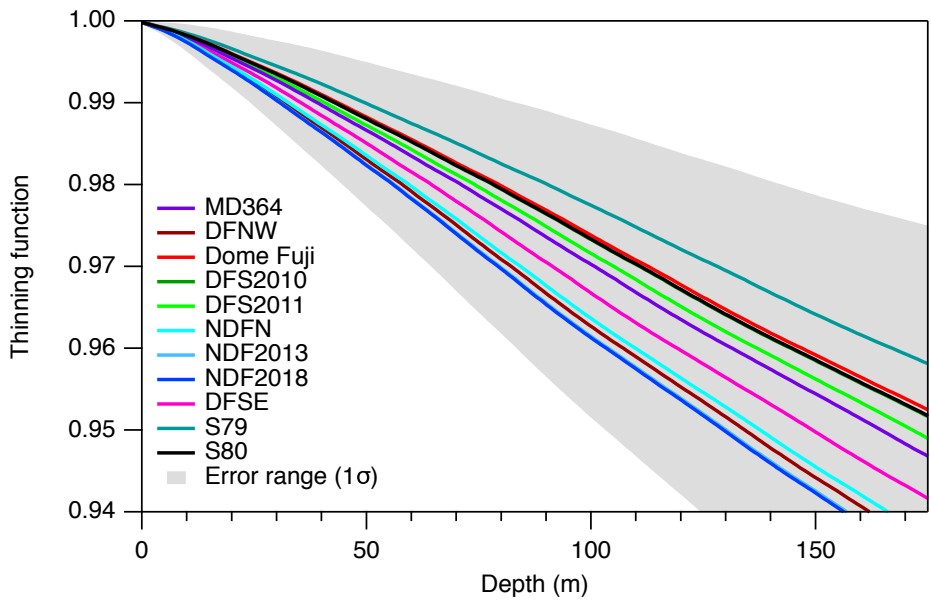


**Figure 5: Thinning function with 1σ uncertainty. The data for Dome Fuji is from Parrenin et al. (2007).**



For DF1, DF2, DFS2011, DFS2010, NDFN and NDF2018 cores covering more than 2 kyr, we calculated the average
accumulation rates over the periods of 0–1850 C.E. and 2000 B.C.E.–1850 C.E. Because there are no volcanic tie points
exactly at 0 C.E. and 2000 B.C.E., we interpolated the volcanic depth-age relationships for the DF1, DF2, NDF, NDFN,
DFS2010 and DFS2011 cores using a probabilistic dating model *Paleochrono* (Parrenin et al., 2021) to derive the depths for
the exact years in the cores. The model estimates the age scale of a core by optimizing the prior estimates of accumulation
rate ($A$) and thinning ($\tau$) as functions of depth to reproduce the chronological constraints. We used the accumulation rate and
thinning function and their uncertainties from the method described above as the prior $A$ and $\tau$. For the age constraints, we
used the age differences between volcanic layers (ice age intervals) constructed from volcanic synchronizations. The
uncertainty of the ice age interval is estimated to be 0.1–1.2 years (one standard deviation) from the following components:
(1) uncertainty in the layer-counted WD2014 age scale (0.1–0.9 years for the intervals, larger for the deeper depth) and (2)
depth uncertainty in matching our cores to WDC (< 1 year, associated with data resolution).

## 360 3. Results

### 3.1 SMB histories from individual cores and snow pits

Figure 6 shows the time series of accumulation rates of all 13 ice cores from 9 sites. The longest record covers 5152 years
(2001 C.E.–3151 B.C.E., DF2 core). The mean accumulation rate over the last 5 kyr is 26.2±1.0 kg m$^{-2}$ yr$^{-1}$ at the Dome Fuji
station (mean of DF1 and DF2). A previously published estimate of the accumulation rate at the Dome Fuji station is 25 and
25.5 kg m$^{-2}$ yr$^{-1}$ over the period 1260–1993 C.E. and 1260–2001 C.E., based on the DF1993 (shallow part of the DF1 core,
Watanabe et al., 1997) and DF2001 (shallow part of the DF2 core, Igarashi et al., 2011), respectively, with fewer age control
points. They agree with our result for the same core and period (24.7±0.1 and 25.5±0.1 kg m$^{-2}$ yr$^{-1}$ for DF1993 and DF2001,
respectively).





**Figure 6: Accumulation rates around Dome Fuji and other high-elevation sites in Dronning Maud Land, reconstructed from shallow ice cores and snow pits. Horizontal black dashed line in each panel indicates the average value at the Dome Fuji station over the last 5 kyr (26.2 kg m⁻² yr⁻¹). Grey shading indicates the estimated uncertainty (2σ). Dotted line indicates a statistically significant long-term trend from the bottom year to 1850 C.E. (to 1461 C.E. for the DF1997). Slope values are summarized in Table 4.**






For the sites within 100 km of the Dome Fuji station, the accumulation rates generally range from ~20 to 30 kg m$^{-2}$ yr$^{-1}$. We note that the variability (periodicity and amplitude) of each record is determined not only by the natural variability but also by the number of age tie points. For example, the DF2 data show higher centennial-scale variability than the DF1 data because of more tie points. We also note that, at around 1000 B.C.E., the DF1 tie points are scarce and the DF1 accumulation

rate is larger than the DF2 value on average over ~600 years. The reason is not clear, but it might suggest possible errors in depth assignments of the ice cores (we note that it is just below the firn-ice transition where the shallow ice-core data are connected to the deep ice-core data for the DF cores). The discrepancy for this part does not affect the following discussion and conclusion about the long-term trends and differences between the sites. In general, the accumulation rate is larger (smaller) to the north (south) of the Dome Fuji station (Fujita et al., 2011). The mean accumulation rate and its variability at

MD364 (range: ~20–60 kg m$^{-2}$ yr$^{-1}$) are much larger than those in the Dome Fuji area. The site is located at the altitude of ~3350 m, lower by ~450 m than at Dome Fuji station, on the north side of the topographic ridge, thus the mean precipitation is expected to be larger than at Dome Fuji. The larger variability is probably caused by horizontal ice flow (4.1 m yr$^{-1}$) over rough bedrock topography, creating the variations in local surface topography (e.g., surface slope and convex/concave) and thus depositional environment (Fujita et al., 2002; Kahle et al., 2021).


For all the individual SMB time series of six cores (DF1, DF2, DFS2010, DFS2011, NDF2018 and NDFN) covering more than 2000 years, we find clear decreasing trends at the rates from -0.05 to -0.09 kg m$^{-2}$ per century (Fig. 6, Table 4). Also, there appears to be a significant increasing trend for the last 200 years for most cores. The increasing trend in the last 200 years is not observed at DF1 (drilled in 1993), possibly due to the lack the record after 1993, when the accumulation rate

might have significantly increased, or the depth assignment might be imprecise for the shallowest part of the core. The increasing trend is also not observed for MD364 with the large temporal variabilities (see above).

To examine the spatial differences in accumulation rate in the inland plateau region around Dome Fuji, mean accumulation rates over fixed time intervals were calculated for each core. For the averaging periods, we chose (1) 1461–1816 C.E. (all

cores), (2) 0–1850 C.E. (DF1, DF2, DFS2010, DFS2011, NDFN and NDF2018 cores), and (3) 2000 B.C.E.–0 C.E. (DF1, DF2, NDFN and NDF2018 cores) (Figure 7). For 1461–1816 C.E., the mean accumulation rates south of the Dome Fuji station (NDFN, NDF2013, NDF2018, S79 and S80), i.e., inland side of the ice divides (Fig. 1), are generally lower than at Dome Fuji (DF1, DF2, DF1997, DFS2011 and DFS2010). On the other hand, the mean accumulation rates north of the Dome Fuji station (MD364 and DFNW) are higher than at Dome Fuji. For example, the accumulation rates at NDFN and

NDF are ~9% and ~13% lower, while that at DFNW is ~8% higher than at Dome Fuji. We find small but significant differences between the DF and DFS sites (~10 km apart) as well as between the NDFN and NDF sites (~5 km apart). These spatial differences are qualitatively consistent with previous studies that the spatial distribution of SMB in the DML region





depends on the elevation, latitude and geographical location of the sites relative to the ice divides (Fujita et al., 2011; Van
Liefferinge et al., 2021).


For 0–1850 C.E. (Fig. 7b), the order of SMB of the DF, DFS, NDFN and NDF sites (from the largest to smallest) is
consistent with those for 1461–1816 C.E (Fig. 7a). The ratios of SMB at NDFN and NDF sites to that at DF are also similar
for all three periods (Fig. 7a-c), suggesting that the relationships of SMB between the sites on multi-centennial or longer
timescales have been stable. The mean SMB at the DF, NDFN and NDF sites for the period 2000 B.C.E–0 C.E. are slightly

larger than those for the period 0–1850 C.E., and significantly larger than those for the period 1461–1816 C.E.

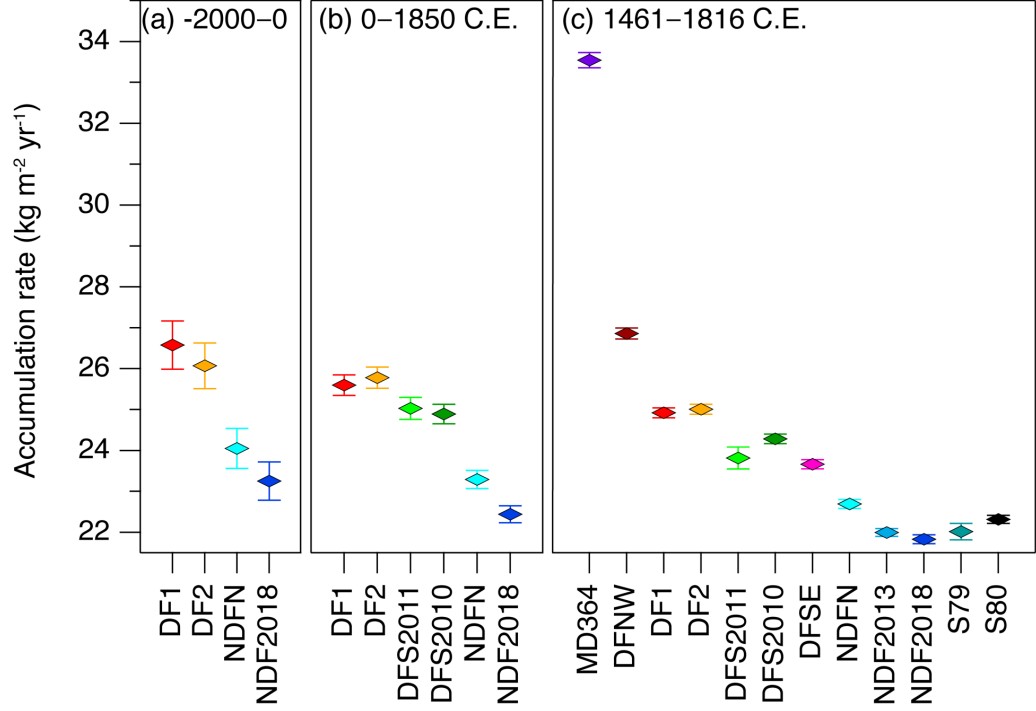

**Figure 7: Mean accumulation rate over three periods: (a) 2000 B.C.E–0 C.E. (b) 0–1850 C.E. (c) 1461–1816 C.E. Error bars**
**represent the 2σ uncertainty of the mean accumulation rate over the respective periods (combined uncertainties in age, density**
**and thinning).**

**Table 3: Mean accumulation rate (kg m$^{-2}$ yr$^{-1}$) over three periods**

| | 1461–1816 C.E. | | 0–1850 C.E. | | 2000 B.C.E.–0 C.E. | |
|---|---|---|---|---|---|---|
| | Ave. | Std. (2σ) | Ave. | Std. (2σ) | Ave. | Std. (2σ) |





| | | | | | | |
|---|---|---|---|---|---|---|
| MD364 | 33.54 | 0.19 | | | | |
| DFNW | 26.86 | 0.14 | | | | |
| DF1 | 24.92 | 0.12 | 26.11 | 0.40 | 26.58 | 0.59 |
| DF2 | 25.01 | 0.12 | 25.78 | 0.26 | 26.07 | 0.56 |
| DFS2011 | 23.81 | 0.27 | 25.03 | 0.27 | | |
| DFS2010 | 24.28 | 0.12 | 24.89 | 0.24 | | |
| DFSE | 23.66 | 0.11 | | | | |
| NDFN | 22.69 | 0.11 | 23.29 | 0.22 | 24.05 | 0.49 |
| NDF2013 | 21.99 | 0.10 | | | | |
| NDF2018 | 21.83 | 0.11 | 22.44 | 0.21 | 23.25 | 0.47 |
| S79 | 22.02 | 0.20 | | | | |
| S80 | 22.32 | 0.10 | | | | |


**Table 4: Temporal trend of accumulation rate**

| | Bottom–1850 C.E. | 0–1850 C.E. | 1000–1850 C.E. | 1850–2019 C.E. |
|---|---|---|---|---|
| | kg m$^{-2}$ 100 yr$^{-1}$ (±1σ) | kg m$^{-2}$ 100 yr$^{-1}$ (±1σ) | kg m$^{-2}$ 100 yr$^{-1}$ (±1σ) | kg m$^{-2}$ 100 yr$^{-1}$ (±1σ) |
| MD364 | +0.380±0.007 (since 436 C.E.) | | -0.280±0.001 | |
| DFNW | -0.326±0.043 (since 1232 C.E.) | | -0.326±0.043 (since 1232 C.E.) | |
| DF1 | -0.042±0.006 (since 2908 B.C.E.) | -0.069 ± 0.001 | -0.152±0.003 | |
| DF1997 | -0.071±0.002 (424 B.C.E.–1460 C.E.) | -0.092 ± 0.003 (0–1460 C.E.) | +0.002±0.030 (1000–1460 C.E.) | |
| DF2 | -0.031±0.006 (since 3151 B.C.E.) | -0.050 ± 0.001 | -0.166±0.004 | |
| DFS2011 | -0.062±0.004 (since 720 B.C.E.) | | -0.246±0.001 | |
| DFS2010 | -0.070±0.003 (since 720 B.C.E.) | | -0.148±0.010 | |
| DFSE | +0.241±0.025 (since 1173 C.E.) | | +0.241±0.025 (since 1173 C.E.) | |
| NDFN | -0.039±0.004 (since 2355 B.C.E.) | -0.060 ± 0.001 | -0.163±0.006 | |
| NDF2013 | -0.028±0.031 (since 1461 C.E.) | | | |
| NDF2018 | -0.043±0.004 | -0.063±0.001 | +0.023±0.007 | |





| | | | | |
|---|---|---|---|---|
| | (since 2714 B.C.E.) | | | |
| S79 | -0.111±0.001 (since 1041 C.E.) | | -0.111±0.001 (since 1041 C.E.) | |
| S80 | -0.165±0.029 (since 1461 C.E.) | | | |
| Stack 1 (unitless) | -0.041±0.001 (since 3151 B.C.E) | -0.105±0.006 | -0.178±0.011 | +2.201±0.560 |
| Stack 2 | -0.037±0.005 (since 3151 B.C.E) | -0.065±0.001 | +0.115±0.002 | +1.308±0.339 |
| 200-yr bin (unitless) (3081 B.C.E …1919 C.E. bin centers) | -0.055 (p < 0.01) | -0.100 (not significant) | -0.199 (not significant) | |
| 200-yr bin (unitless) (2900 B.C.E …1900 C.E. bin centers) | -0.057 (p < 0.01) | -0.127 (p < 0.01) | -0.167 (not significant) | |
| 200-yr bin (unitless) (2950 B.C.E …1850 C.E. bin centers) | -0.050 (p < 0.01) | -0.113 (p < 0.01) | -0.190 (not significant) | |
| 200-yr bin (unitless) (3000 B.C.E …1800 C.E. bin centers) | -0.051 (p < 0.01) | -0.107 (p < 0.01) | -0.196 (p < 0.01) | |
| DF stack (DF1, DF2, DF1997) | -0.040±0.006 (since 3151 B.C.E.) | -0.061±0.001 | -0.142±0.002 | |
| DFS stack (DFS2010, DFS2011) | -0.064±0.004 (since 720 B.C.E.) | -0.080±0.002 | -0.195±0.004 | |
| NDF-NDFN stack (NDF2013, NDF2018, NDFN) | -0.039±0.005 (since 2714 B.C.E.) | -0.065±0.002 | -0.069±0.003 | |


The recent accumulation rates estimated from snow pit observations are shown in Table 5. The average accumulation rate of all pit data since 1992 C.E. is 24.8±1.0 kg m$^{-2}$ yr$^{-1}$, which is in general agreement with previously reported values over similar periods from a pit study (25.6 kg m$^{-2}$ yr$^{-1}$ for 1992–2007 C.E., Hoshina et al., 2014) and a snow stake study (27.3±1.5 kg m$^{-2}$ yr$^{-1}$ for 1995–2006 C.E., Kameda et al., 2008). However, there are no consistent spatial gradients relative to the topographic ridge (i.e., less accumulation on the inland side of the ridge) for the average values since 1992 C.E. as found for the long-term reconstructions from the ice cores. The large variability of density near the surface and redistribution of surface snow (intermittent removal, Kameda et al., 2008) may introduce relatively large uncertainty in the estimates. On the other hand, the mean accumulation rates at MD732 (close to the DF station) since 1966 C.E. and 1964 C.E. are significantly larger than those at NDF, which is consistent with the spatial gradient found from the ice cores. The accumulation rates at MD732 and NDF since 1964 C.E. are significantly larger than the average values over the last 4 kyr and comparable to the largest values about 5 kyr ago. Although precise comparisons may be difficult because of the different averaging periods (pit and ice core records average about ~50 years and a few centuries, respectively), the accumulation rate around Dome Fuji in the last few decades appears to be high in the long-term perspective.


**Table 5: Accumulation rate for the last few decades from snow pit data**





| | Observation date (yyyy/mm/dd) | Surface to Pinatubo (1992) kg m⁻² yr⁻¹ | Std. (1σ) | Surface to Bomb test (1966) kg m⁻² yr⁻¹ | Std. (1σ) | Surface to Agung (1964) kg m⁻² yr⁻¹ | Std. (1σ) |
|---|---|---|---|---|---|---|---|
| MD732 | 2007/12/10–11 | 24.0 | 2.9 | 26.8 | 1.3 | 28.3 | 1.2 |
| Dome Fuji | 2003/2/4 | | | 29.0 | 1.7 | 32.1 | 1.6 |
| DFS2010 | 2010/1/22 | 24.0 | 2.5 | | | | |
| NDF2013 | 2012/12/22 | 25.1 | 2.2 | | | | |
| NDF2018 | 2017/12/27–29 | 24.7 | 1.8 | 25.5 | 0.8 | 24.7 | 0.8 |
| S79 | 2013/1/5 | 25.1 | 2.2 | | | | |
| S80 | 2012/12/30 | 25.8 | 2.3 | | | | |

## 3.2 Stacked record

To robustly estimate the trends and variability of SMB in the Dome Fuji area (Fig. 1), we stack the reconstructed SMB from the individual cores. We find that the modern SMB at the study sites are within ±10 % of that at Dome Fuji, and it is not known if the variability of SMB is a function of its mean value. Thus, we stacked the records in two methods, with or without normalizing the variability. We excluded the DF1997 core from the stacking because it lacks reliable SMB reconstruction for the last ~550 years.


In the first stacking method (Stack 1), each of the 12 SMB time series from 9 sites is normalized to its average and one standard deviation over the period 1461–1816 C.E., and 12 records are simply averaged. The stacked record is smoothed with a 71-year moving average to reduce short-term noise. In the second stacking method (Stack 2), each of the 9 SMB time series from within 100 km of the Dome Fuji is normalized to its average between 1461 C.E. and 1816 C.E. and smoothed

with a 31-year moving average. The smoothed records are averaged, added by the average SMB of the 9 sites (23.90 kg m⁻² yr⁻¹ for 1461–1816 C.E.), and smoothed with a 71-year moving average. We exclude S79, S80 and MD364 records, which are located more than 100 km away from the Dome Fuji station. The advantage of this method is to permit a quantitative reconstruction, and it is necessary to exclude MD364 because its average and variability are significantly larger than those near Dome Fuji. The S79 and S80 have similar accumulation rates to Dome Fuji, but their locations are clustered in one

direction relative to the dome (Fig. 1). The uncertainties of Stack 1 and Stack 2 were estimated by a Monte Carlo approach, in which the SMB value of each core in each age segment (between the age control points) was randomly varied 1000 times according to the error estimate and stacked. The uncertainties of the long-term trends of Stack 1 and Stack 2 were also estimated as the standard deviation of the slope of regression lines to the same 1000 stacked records by the Monte Carlo method.


The mean SMB of stack 2 is 25.2±0.8 kg m⁻² yr⁻¹ over the entire period. The centennial-scale variabilities in both stacks are very similar to each other (Fig. 8a, b), although the numbers of stacked data are different between the two methods for the





last 1500 years. There are general long-term declining trends in both stacked records over the entire period (~5000 years), with a minimum accumulation rate around 1700–1800 C.E. and the greatest increase during the last 150 years.


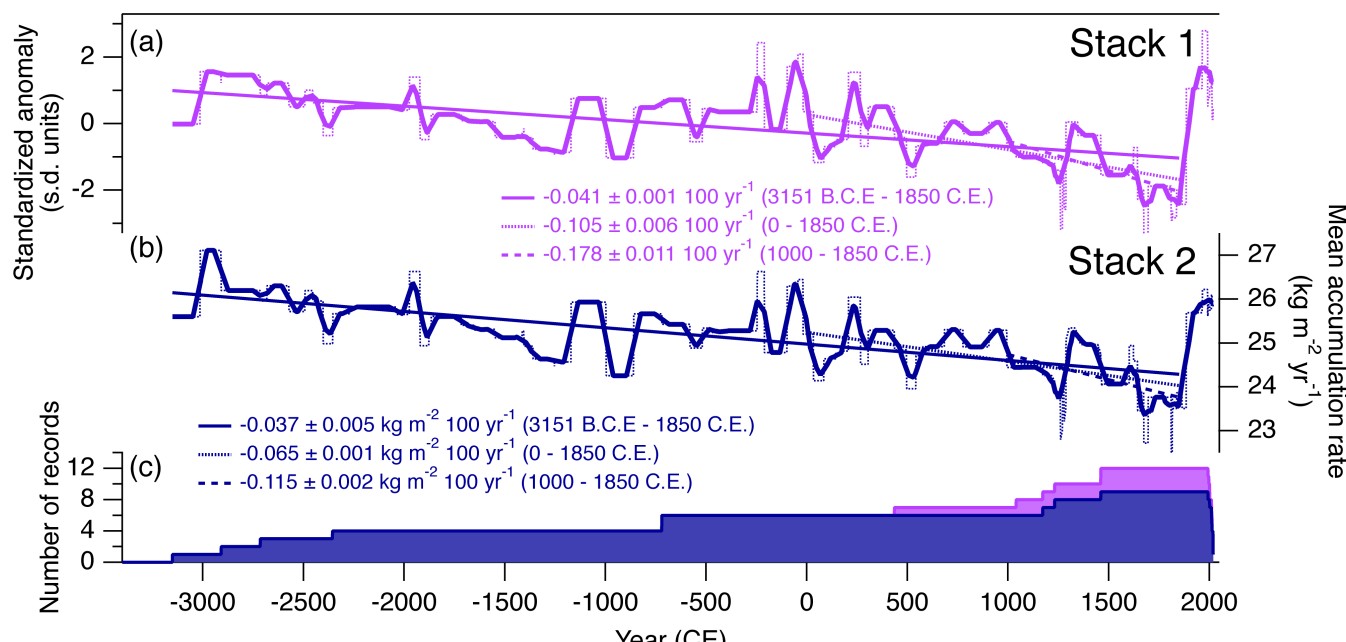

**Figure 8: Composites of accumulation rates over the last 5000 years after (a) normalizing using both the mean and standard deviation of each core (Stack 1), and (b) normalizing only using the mean of each core near Dome Fuji (Stack 2), and (c) the number of records that contribute to the composite. See text for details of stacking methods.**


Both stacked records show statistically significant decreasing trends over the most recent ~5000 years (1850 C.E.–3151 B.C.E.), with the slope of -0.037±0.005 kg m$^{-2}$ per century for Stack 2. We analyze the robustness of the long-term trend following the method of McGregor et al. (2015), by calculating the average accumulation rate over 200-yr intervals (bins)

for each core, taking the median value in each bin, making the composite record by connecting the median values, and drawing the regression line (Fig. 9). We also shift the position of the bin in 50-yr increments to assess its influence on the regression slope (McGregor et al., 2015). All binned records show similar decreasing trends over the 5000 years, indicating that the decreasing accumulation rate is a robust feature around Dome Fuji. The slopes from the binned reconstructions are smaller than that of Stack 1, probably because the binning method reconstructs a lower accumulation rate from ~1000 B.C.E

to ~1000 C.E. than Stack 1 by taking the median values rather than the average values. The long-term trends for the periods of 0 to 1850 C.E. (Table 4) for Stack1 and Stack 2, as well as all the binned records, also show negative trends (Fig. 8 and 9).




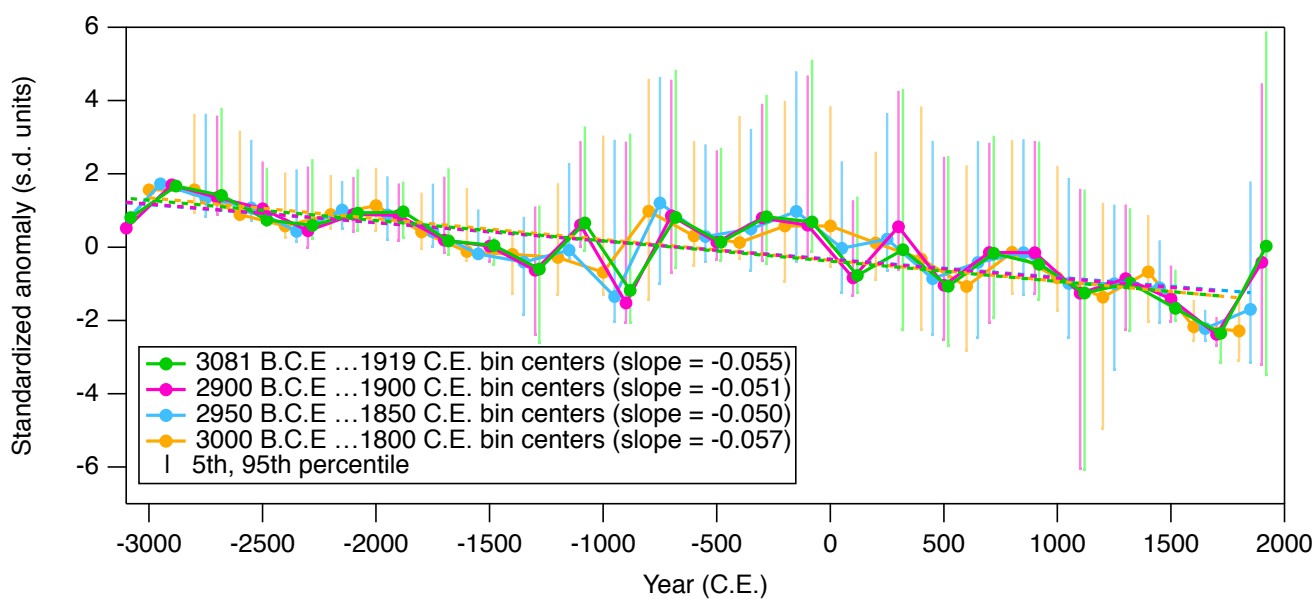

**Figure 9: SMB composites by connecting medians in consecutive 200-yr bins. Different colors indicate different ranges of bins (offsets by ~50 years). Vertical bar indicates 5–95th percentile. Dotted line indicates a statistically significant long-term trend.**

We also made three local stacks using the subsets of data from sites close to each other, as follows; "DF stack" with DF1, DF2 and DF1997 cores, "DFS stack" with DFS2010 and DFS2011 cores and "NDF-NDFN stack" with NDFN, NDF2013 and NDF2018 cores (Fig. 10). For the last 2700 years, the local SMB stacks show systematic differences (DF > DFS > NDF-NDFN) with similar centennial-scale variations. We also find significant decreasing trends in all local stacks, with the slope of the DFS stack being slightly steeper than the other two stacks. This might be related to possible long-term change of local

accumulation environments due to migration of dome summit position in combination with horizontal ice flow (Saito, 2002; Parrenin et al., 2016).





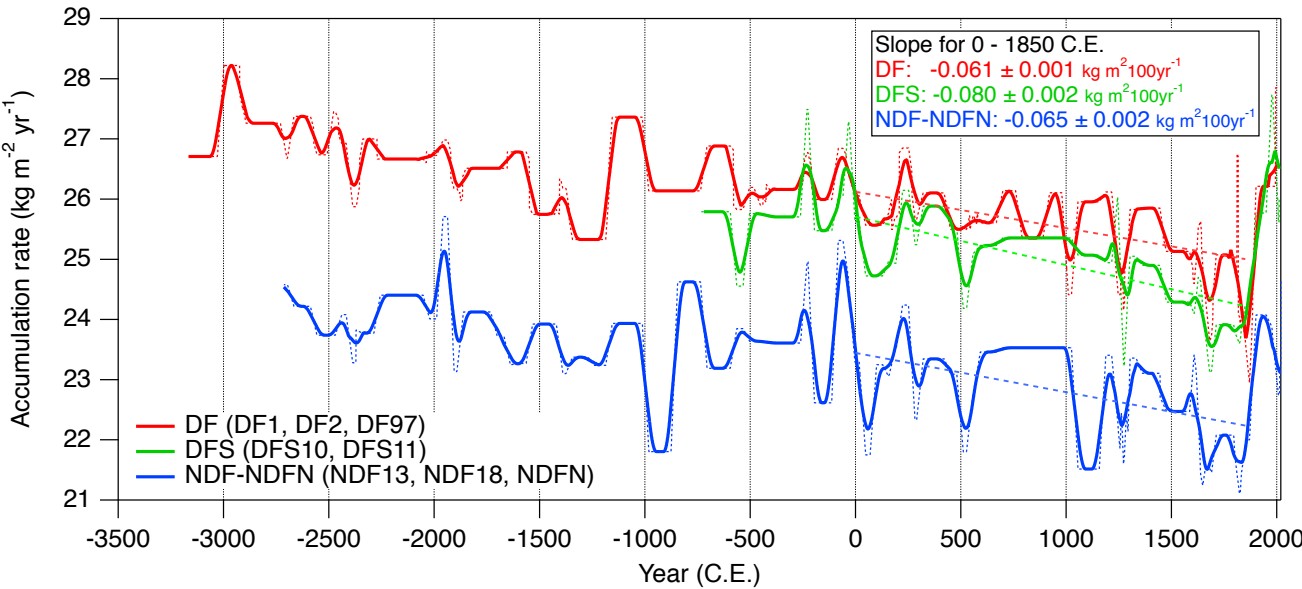

**Figure 10: SMB composites for the DF, DFS and NDF regions. The DF stack was calculated as a simple average of the SMB data of the DF1, DF2 and DF1997 cores after smoothing each record with a 31-year moving average (dotted line). Solid line indicates the SMB stack with a 71-year moving average. The DFS and NDF stacks were calculated in the same manner as the DF stack using DFS2010 and DFS2011 for the DFS stack and NDF2013, NDF2018 and NDFN for the NDF stack, respectively. Dashed line indicates a statistically significant trend for 0–1850 C.E.**

## 4. Discussion

### 4.1 Multi-millennial trend

We first discuss the robustness of the long-term trends. The long-term decreasing trend in the DF area is largely determined by the six ice cores going back by more than 2 kyr (DF1, DF2, DFS2011, DFS2010, NDFN and NDF2018). As the depths of these cores are more than 100 m, the corrections for layer thinning due to ice flow impact the reconstructed SMB records. Because the thinning is weakly constrained for the shallow cores (except for the DF core with good age control from the deeper depths), we investigate the possibility of reversing the sign of trend due to errors in thinning. In particular, we assess whether the negative accumulation rate trend is possibly an artifact of thinning correction (section 2.5), because the correction increases the estimated mass (hence SMB) at deeper depths. Here, we calculate the SMB under the (unrealistic) assumption of no layer thinning for the six cores. As shown in Figure 11, all cores exhibit a significant negative accumulation rate trend even without the thinning correction. Therefore, we conclude that the multi-millennial decreasing trend in accumulation is a robust feature in the Dome Fuji area.


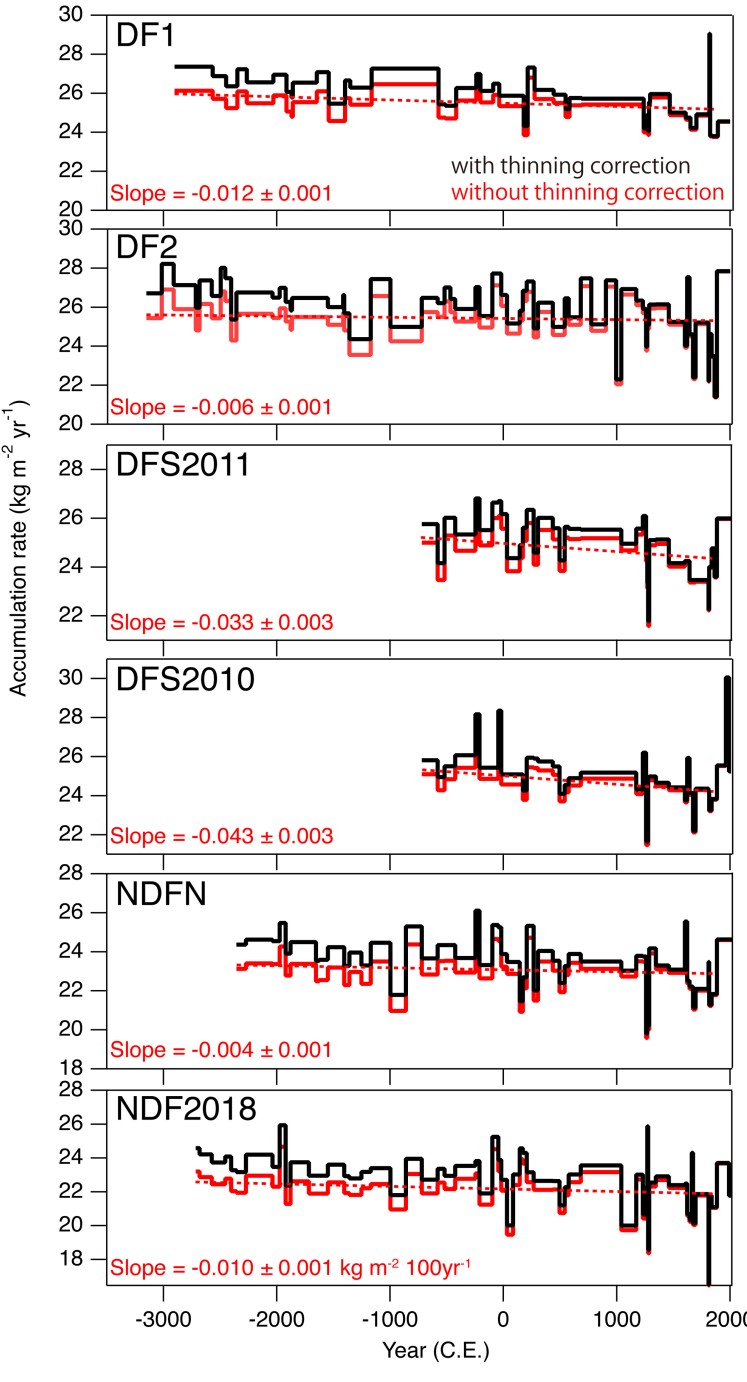

**Figure 11: Temporal changes in accumulation rate with and without thinning correction. Statistically significant slope values from the results without thinning correction are indicated.**




We discuss possible causes for the decreasing trend in accumulation rate around Dome Fuji over the last ~5000 years in the preindustrial period. Note that the main objective of this study is a reliable reconstruction of SMB over long timescales. The thorough investigation of the mechanisms would require reliable SMB reconstructions in other parts of Antarctica as well as

climate modeling works, which are beyond the scope of this study. The first obvious candidate for the long-term change in accumulation rate is the secular changes in orbital forcings. Obliquity (the tilt of Earth's rotation axis) becomes smaller over the last 5000 years (Fig. 12c), which gradually decreases the annual mean insolation at high latitudes. The declining annual mean insolation may cool the ocean surface and atmosphere over the Southern Ocean and Antarctica, leading to reduced evaporation and atmospheric moisture content, and possibly the general decrease in the snow accumulation over the

Antarctic inland. On the other hand, the radiative forcing of greenhouse gases increases over the same period (Fig. 12d), which might counteract other forcings for reducing snow accumulation. Other potentially important forcings are the changes in solar irradiance (Fig. 12e) as well as volcanic forcing (Fig. 12f), although they have significant uncertainties in reconstructions.





**Figure 12: Dome Fuji SMB record and climatic forcings over the last 5000 years. (a) Dome Fuji stacked SMB (Stack 2), (b) detrended SMB of Stack 1, (c) obliquity, (d) radiative forcing (RF) of greenhouse gases (CO$_2$, CH$_4$ and N$_2$O) relative to 1750 C.E. calculated using CO$_2$, CH$_4$ and N$_2$O concentrations from Bereiter et al. (2015), Buizert et al. (2015), Fischer et al. (2019), respectively, and equations proposed by Etminan et al. (2016), (e) RF of total solar irradiance (Wu et al., 2018) relative to 1750 C.E., (f) centennial mean volcanic RF at 40–90˚S, and (g) stratospheric sulfur injections from volcanic eruptions (TgS = teragram of sulfur) (Sigl et al., 2022). Volcanic RF was calculated by stratospheric aerosol optical depth (Sigl et al., 2022) and a conversion factor (Hansen et al., 2005).**



For the AIS SMB, sensitivity experiments with a regional climate model have suggested the roles of Southern Ocean sea-surface temperature (SST) and sea-ice concentration for the SMB (Kittel et al., 2018). Their results showed that increases in sea-ice concentration led to a decrease in the SMB over most of the AIS, but lower SST led to contrasting changes in the

coastal and inland areas of AIS, i.e., decrease in coastal areas and increase in inland, because of the relationship between atmospheric vapor content and saturation pressure along the air-mass trajectory towards the inland. On the other hand, Vannitsem et al. (2019) analyzed the relationship between reconstructed AIS SMB (Thomas et al., 2017) and the results of global climate models for 850–2005 C.E. and found that the SMB over the Antarctic Plateau is mostly influenced by the surface air temperature and sea ice concentration, and not by large-scale atmospheric modes such as El Niño or Southern

Annular Mode (SAM). Therefore, to provide data-based clues for the long-term relationships between the surface temperature, sea ice and SMB, we compare our SMB trend with the published temperature and sea ice reconstructions over the latter half of the Holocene.

Long-term coolings from the mid- to late-Holocene at high latitudes in the Southern Hemisphere have been suggested by

previous studies, although the relevant climatic forcings and mechanisms (e.g., in terms of orbital configurations and greenhouse gases) are not clear. Recent compilations of surface temperature reconstructions exhibit cooling trends for the mid to high latitudes of the Southern Hemisphere (Marcott et al., 2013; Kaufman et al., 2020a; Kaufman et al., 2020b), and stable water isotope records on the EAP (EDML, Dome Fuji, Vostok, Dome C, TALDICE) commonly show decreasing trend from the mid-to late-Holocene (Masson-Delmotte et al., 2011). On the other hand, individual reconstructions of sea

surface temperature (SST) show a variety of trends from clear cooling to little trend over the last 5000 years (e.g., Hodell et al., 2001; Nielsen et al., 2004; Anderson et al., 2009; Divine et al., 2010; Lamy et al., 2010; Shevenell et al., 2011; Etourneau et al., 2013; Xiao et al., 2016), which may be partly because local SST may be sensitive to the positions relative to ocean currents. Cooling on land in the latter half of the Holocene is also suggested from the advances of glaciers in the Southern Hemisphere (Solomina et al., 2015).


Reconstructed sea ice extents from multiple proxy records for different parts of the Southern Ocean show general advancements for the last 5000 years. The sea ice advance around 5000 to 4000 years ago was found from TN057-13 in the southeast Atlantic (Hoddel et al., 2001), from JPC24 in Prydz bay in the Atlantic Ocean sector (Denis et al., 2010), and from MD03-2601 (Crosta et al., 2008; Denis et al., 2010) and U1357 (Ashley et al., 2021) offshore Adélie Land in the Indian

Ocean sector. Denis et al. (2010) suggested that increasing sea ice cover over the late Holocene is a common feature in the coastal Antarctic. From the Antarctic ice cores, secular increases in sodium, which is a proxy for sea ice extent in the Southern Ocean (Wolff et al., 2003), during the Holocene have been widely observed (Winski et al., 2021). From ice cores in the EAP, the increases in sodium fluxes/concentrations since mid-Holocene were observed at EDML (Fischer et al., 2007), Dome Fuji (Iizuka et al., 2008), Dome C (Fischer et al., 2007), and TALDICE (Mezgec et al., 2017). Thus, sea ice extent

was probably increased in the areas of the Southern Ocean that supply sodium to the EAP.





From the above evidence, the decreasing trend in the Dome Fuji SMB over the last 5000 years appears to be associated with surface cooling and sea-ice expansion in the potential water vapor source areas for the Dome Fuji region, namely the Atlantic Ocean and Indian Ocean sectors of Southern Ocean (Suzuki et al., 2008). The cooling trend is also seen over the

EAP, possibly suggesting a general cooling trend of the atmosphere from the vapor source regions to the EAP. Our observation is consistent with the analyses of Vannitsem et al. (2019) that both surface air temperature and sea ice trends could have contributed to the decreasing Dome Fuji SMB. On the contrary, our observation may be inconsistent with the sensitivity study of the regional climate model (Kittel et al., 2018) that the decrease of accumulation rate in the EAP requires an increase in SST. A thorough investigation of the inconsistency is beyond the scope of this study, but it might be related to

the differences in the background climate states between the model runs (1979–2015) and the past 5000 years.

## 4.2 Centennial-scale variability

We compare the centennial-scale variabilities of SMB records at Dome Fuji and other parts of Antarctica for the last 1000 years. For Dome Fuji, the deviation of SMB from the long-term trend was calculated by subtracting a linear function fitted

through 1236–1850 C.E. (the oldest year for the fitting was adopted from the other records with a shorter duration). For the comparison, we use the SMB reconstruction for the Atlantic sector of DML based on the analyses of six firn cores (Hofstede et al., 2004), as well as the composite SMB records for the EAP, whole East Antarctica (EA; including coastal area of DML, Wilkes Land and East Antarctic Plateau), and West Antarctica (WA; Antarctic Peninsula is not included) based on the compilations of multiple ice cores (Thomas et al., 2017). The published records by Thomas et al. (2017) were smoothed by a

71-yr moving average to match the resolution with that of the Dome Fuji data, and the deviations from their linear trends were calculated in the same manner as the Dome Fuji record. Figure 13 shows the SMB anomaly for the Dome Fuji, Atlantic sector of DML, EAP, EA, and AIS. The SMB anomaly around Dome Fuji has four distinct periods: mostly negative before 1300 C.E., slightly positive for 1300–1450 C.E., slightly negative for 1450–1850 C.E., and positive after 1850 C.E. (Fig. 13a). Before 1460 C.E., the Dome Fuji, DML, EAP and EA records commonly exhibit a peak around 1300–1450 C.E.,

suggesting this anomaly is a real climatic signal and extended to much of East Antarctica. For the middle part (1460–1850 C.E.), the DML, EAP and EA records are also characterized by a broad minimum around 1500 C.E. and a relatively short maximum around 1600 C.E. The Dome Fuji record show consistently negative values in this period, but it also has a broad minimum around 1500 C.E. and small maximum around 1600 C.E. The WA composite also show a positive anomaly about 1600 C.E. After ~1650 C.E., Dome Fuji, DML, EAP, EA and WA records all show a minimum around 1700 C.E., although

the durations of the negative anomalies are different. The Dome Fuji and DML records show minima in more recent periods (around 1850 C.E. in the Dome Fuji and around 1900 C.E. in the DML), which are not seen in the other records. For the most recent part, all records show increases with different durations (over 100–300 years).



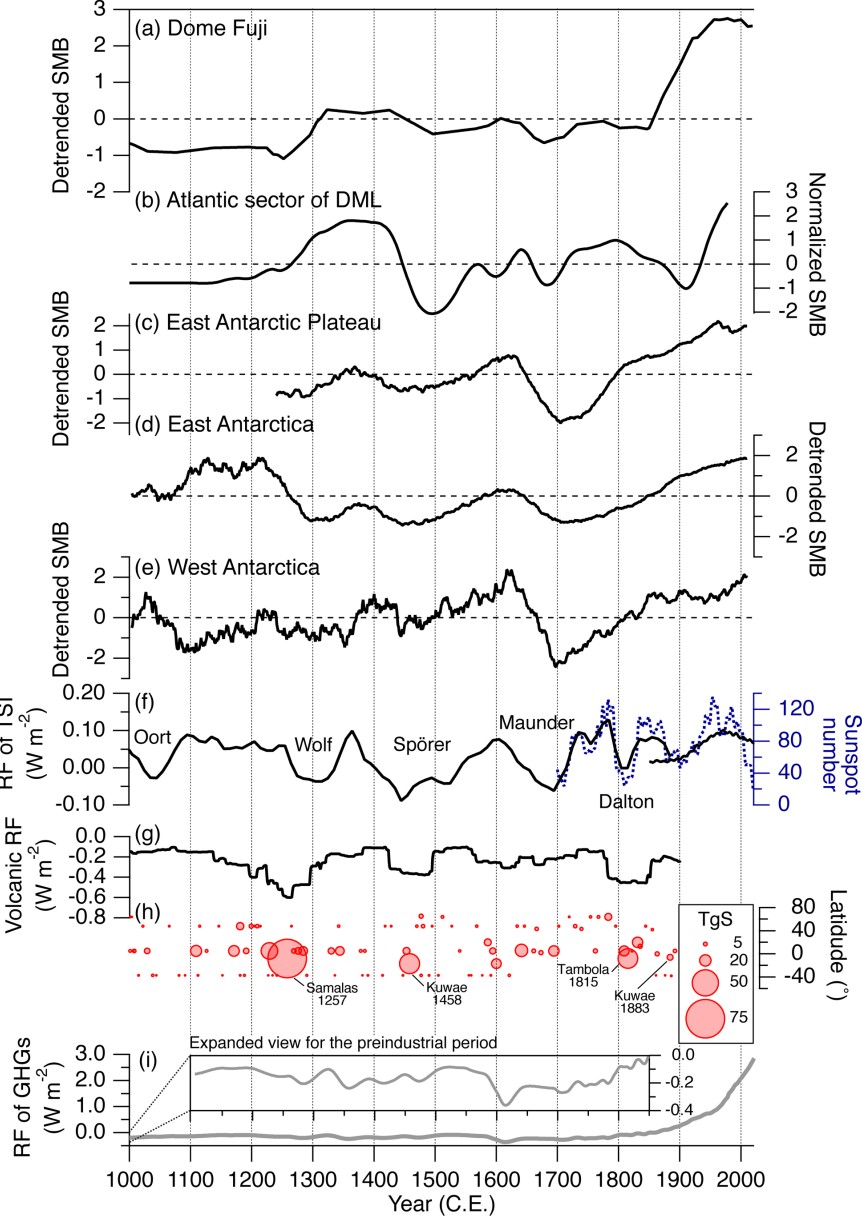

**Figure 13: Detrended SMB for the last 1000 years for (a) Dome Fuji, (b) Atlantic sector of DML (Hofsted et al., 2004), (c) East Antarctic Plateau (detrended EAP SMB of Thomas et al., 2017), (d) East Antarctica (detrended EA SMB of Thomas et al., 2017), and (e) West Antarctica (detrended WA SMB of Thomas et al., 2017). Also shown are (f) radiative forcing (RF) of total solar irradiance relative to 1750 C.E. (black, Wu et al., 2018) and sunspot number, (g) mean volcanic RF at 40–90°S with a 71-year moving average, (h) stratospheric sulfur injections from volcanic eruptions (Sigl et al., 2022), and (i) RF of greenhouse gases (CO₂, CH₄ and N₂O) relative to 1750 C.E. TSI (1000–1885 C.E.) and sunspot number data (1700–2021 C.E.) are from Max Planck Institute for Solar System Research "Solar Variability and Climate" group, https://doi.org/10.17617/1.5U, and the World Data Center SILSO, Royal Observatory of Belgium, Brussels, https://www.sidc.be/silso/datafiles). TSI of 1850–2020 C.E. is from Matthes et al. (2017). Volcanic RF was calculated by stratospheric aerosol optical depth (Sigl et al., 2022) and a conversion factor (Hansen et al., 2005). RF of GHGs was calculated using CO₂, CH₄ and N₂O concentrations from Rubino et al. (2019) and equations proposed by Etminan et al. (2016).**




We further compare our Dome Fuji record with other published reconstructions. In the Princess Elizabeth Land, the SMB anomaly is relatively high for ~1207–1450 C.E., low for ~1450–1850 C.E. and significantly increased afterward (until 1996 C.E.) (Li et al., 2009). In the South Pole record (Ferris et al., 2011), the SMB anomaly is relatively large for ~1200–1400 C.E., reduced during the period of 1500–1900 C.E., and then increased afterward. On the continental scale, a stacked SMB record by Frezzotti et al. (2013) for the last 800 years, which includes more data from the Antarctic Plateau than the composite of Thomas et al. (2017), identified three periods of low accumulation rate (1250–1300, 1420–1550 and 1660–1790 C.E.). In summary, the reconstructions for the EAP commonly show a higher accumulation rate for the periods of ~1250–1400 and ~1550–1650 C.E. and a lower accumulation rate around 1500 and 1700 C.E. The increases in the snow accumulation over the 20th century are also observed, although the timing of the onsets is different among the records.

Multidecadal to centennial SMB variabilities may be driven by external forcings such as solar irradiance and volcanic activity, as well as by internal variabilities of the atmosphere-ocean system (Goosse et al., 2012; Frezzotti et al., 2013; PAGES 2k Consortium, 2013, 2019; Medley and Thomas, 2019; Mann et al., 2021). Previous studies have suggested the correspondence between the periods of reduced SMB and strong volcanic forcings or solar minima over the last millennium (Bertler et al., 2011; Frezzotti et al., 2013; Osipov et al., 2014; Thomas et al., 2017), possibly through the reduction of incoming solar radiation and associated changes in the climate system (e.g., surface cooling and circulation change). In the detrended SMB record at Dome Fuji over the last 1000 years, the intervals of small SMB anomalies appear to correspond to those with large volcanic eruptions (e.g., Samalas at 1257 C.E., Kuwae at 1458 C.E., Tambora at 1815 C.E. and Krakatau at 1883 C.E.), as well as the Spörer, Maunder and Dalton grand solar minima (Fig. 13). On the other hand, the Oort and Wolf grand solar minima do not correspond to low SMB anomaly periods at Dome Fuji but they coincide with the low SMB anomaly periods in the EA composite. Thus, our data is partly consistent with the previous suggestions for the last 1000 years. As discussed above, the Dome Fuji stacked record shows a negative SMB anomaly for the 15th to 19th century. During this period, in addition to strong volcanic forcing and weak solar activity, the radiative forcing of greenhouse gases becomes the smallest in the last 2 kyr, which may have contributed to the lower SMB.

For the older period, the detrended SMB for 1500 B.C.E.–1000 C.E. shows variability similar to the last 1000 years, and it shows reduced variability before 1500 B.C.E. (Fig. 12). Also, the correspondence between the intervals of low SMB anomalies and strong volcanic or weak solar forcings is not clear in the older period. Before 1000 C.E., the average solar activity was larger and the frequency of large volcanic eruptions was lower than during the last 1000 years. It might be possible that the clear relationship between the significant negative SMB anomalies and solar and volcanic forcings is only visible when the two forcing anomalies are strong and coincide, which seem rarer in the older period than in the last 1000 years. However, there is also a possibility that the SMB variability on the multidecadal to centennial scales is not well





captured by our stack because of the fewer number of ice cores and age constraints. The reconstructions of volcanic and solar
forcings might also have larger uncertainties in the older part. In any case, it is desirable to obtain long-term and detailed
reconstructions of the AIS SMB at multiple sites to examine their relationships with the climatic forcings.

The Dome Fuji SMB record reveals that the magnitude of the snow accumulation increase during the last 150 years is the
greatest in the last 5000 years (+1.308±0.339 kg m$^{-2}$ 100yr$^{-1}$, Table 4). We speculate that the large increase may have been
created by the combination of anomalously low accumulation in the 18–19th century (23.8±0.4 kg m$^{-2}$ yr$^{-1}$ in Stack 2) and a
continuous increase in the industrial period. As many studies have suggested, the significant increase in accumulation rate in
the 20th century may be attributed to anthropogenic forcings such as increased atmospheric greenhouse-gas concentrations
and stratospheric ozone depletion, which would lead to an increase in atmospheric moisture content over the Southern Ocean
and Antarctic continent (Lenaerts et al., 2018; Medley and Thomas, 2019; Chemke et al., 2020; Dalaiden et al., 2020). In the
pre-industrial period, the variation in the combined radiative forcing of solar activity, volcanic forcing and greenhouse gases
is smaller than 1 W m$^{-2}$, while the anthropogenic radiative forcing alone is +2.7 W m$^{-2}$ in the industrial period (in 2019
relative to 1750) (IPCC, 2021). Although the current accumulation rate is likely lower than the maximum value in the last
5000 years of the preindustrial period, we speculate that it could eventually exceed the natural range in the future as the
anthropogenic forcings continue to change the atmosphere-ocean system around Antarctica.

## 5. Conclusion

We analyzed a total of 15 ice cores and 7 snow pit samples to obtain 13 SMB records around Dome Fuji over the last 5000
years. Four ice cores cover more than 4000 years, three ice cores cover 1500 years, six ice cores cover 800 years or less, and
the longest reconstruction covers 5152 years (3151 B.C.E.–2001 C.E., DF2 core). Our new SMB reconstructions took
advantage of detailed and precise depth-age controls for the ice cores owing to high-resolution volcanic synchronization with
the WAIS Divide ice core. We also considered vertical ice thinning for all cores, and used high-resolution permittivity data
for estimating high-precision density profiles for most of the cores. Because the accumulation rate is low and its variability is
relatively large in the high-elevation plateau in East Antarctica (e.g., above 3,000 m a.s.l.), we stacked all available SMB
records from the individual ice cores and snow pits for a reliable reconstruction of the SMB history in the Dome Fuji area.
Our main findings are summarised as follows.


-   The mean accumulation rate over the last 5000 years is 26.2±1.0 kg m$^{-2}$ yr$^{-1}$ at Dome Fuji. The mean accumulation rates
    for the last 4000 years in the preindustrial period are lower (higher) south (north) of the Dome Fuji station. Such spatial
    gradient around Dome Fuji is consistent with the modern observations and centennial-scale reconstructions that it depends
    on the site location relative to the ice ridges combined with prevailing wind directions and proximity to the ocean.




- A statistically significant long-term decreasing trend over the last 5000 years in the preindustrial period is found in the stacked SMB record, with a slope of -0.037±0.005 kg m⁻² per century. We speculate that the long-term trend is attributable to long-term surface cooling over the Southern Ocean and East Antarctica and sea-ice expansion in the moisture source areas.


- After removing the long-term trend, the stacked SMB shows centennial-scale variations. For 0–1850 C.E., the detrended SMB anomaly is mostly negative before 1300 C.E., slightly positive for 1300–1450 C.E., slightly negative for 1450–1850 C.E with a weak maximum around 1600 C.E., and positive after 1850 C.E. with a strong increase. These variations are generally consistent with previous SMB reconstructions in the East Antarctic Plateau, which may be driven by the

combination of strong volcanic forcings and solar minima.

- For the older period, the detrended SMB anomaly for 1500 B.C.E.–1000 C.E. shows variability similar to the last 1000 years, and it shows reduced variability before 1500 B.C.E. The correspondence between the SMB anomalies and climatic forcings is not clear as in the last 1000 years, possibly because of generally larger solar forcings, the lack of coincidence of volcanic and solar forcings, or the deterioration of the SMB reconstruction due to the smaller number of ice cores and

age constraints.

- The magnitude of the increase in accumulation rate during the last 150 years (+1.308±0.339 kg m⁻² per century) is the greatest in the last 5000 years, which may have been forced by the combination of the anomalously low accumulation in

the 18–19th century and anthropogenically forced atmospheric greenhouse-gas increase and stratospheric ozone depletion.

We demonstrated the possibility of detailed and precise age control of low-accumulation ice cores thanks to the synchronization to the layer-counted WAIS Divide core, which may be applied to low-accumulation ice cores from other high-elevation, vast areas of East Antarctica. To further examine the long-term SMB trend as well as centennial scale SMB

variations in relation to the climatic forcings, high-resolution and precise SMB reconstructions with a long time scale from multiple sites are desired.

**Appendix A: Conversion of relative permittivity to density for NDF2018, DFNW, DFSE and NDFN cores**

We constructed a conversion equation from the permittivity measured after 2018 to density. The measured bulk densities of

the NDF2018, DFNW, DFSE, and NDFN cores, as well as the NDF2018 snow pit were used for the conversion. The permittivity data was resampled at 0.5 m intervals to match the resolution of bulk density data of the cores, and a third-order polynomial function was used to represent the permittivity-density relationship:



$\rho = -20.15\varepsilon_h{}^3 - 99.801\varepsilon_h{}^2 + 243.02\varepsilon_h - 220.57$ (kg m$^{-3}$),

where $\rho$ is density (kg m$^{-3}$) and $\varepsilon$ is permittivity (Fig. A1).


Uncertainty for the permittivity-based density is derived from the uncertainties of (1) bulk density measurement and (2) permittivity for a given density. The first component dominates the uncertainty in the relatively shallow part due to the irregular shapes of the core pieces, and the second component dominates the deep part where the bulk density is rather precise. To estimate the depth-dependent density uncertainty consisting of the two components, we employ a Monte Carlo

approach, in which the bulk density and permittivity data were randomly modified 1000 times according to the respective uncertainties and fitted using the polynomial function. The range of density converted from permittivity using the 1000 pseudo datasets gives the uncertainty estimate. An example of the depth profile of density uncertainty is shown in Figure A2 (NDF2018 core).

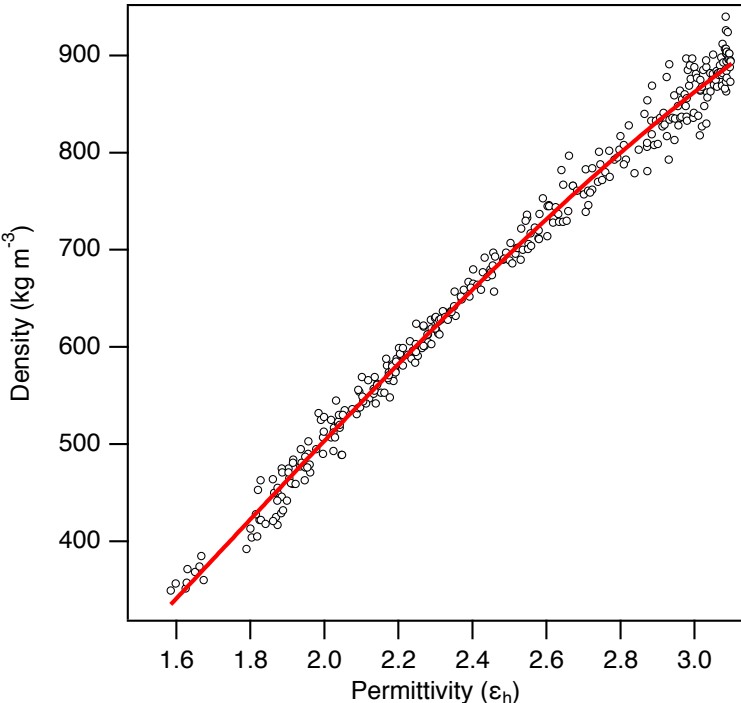


**Figure A1: Scatter plot of the permittivity and bulk density and its third order polynomial fitting curve ($\rho = -20.15\varepsilon^3 + 99.801\varepsilon^2 + 243.02\varepsilon - 220.57$, where $\rho$ is density and $\varepsilon$ is permittivity).**



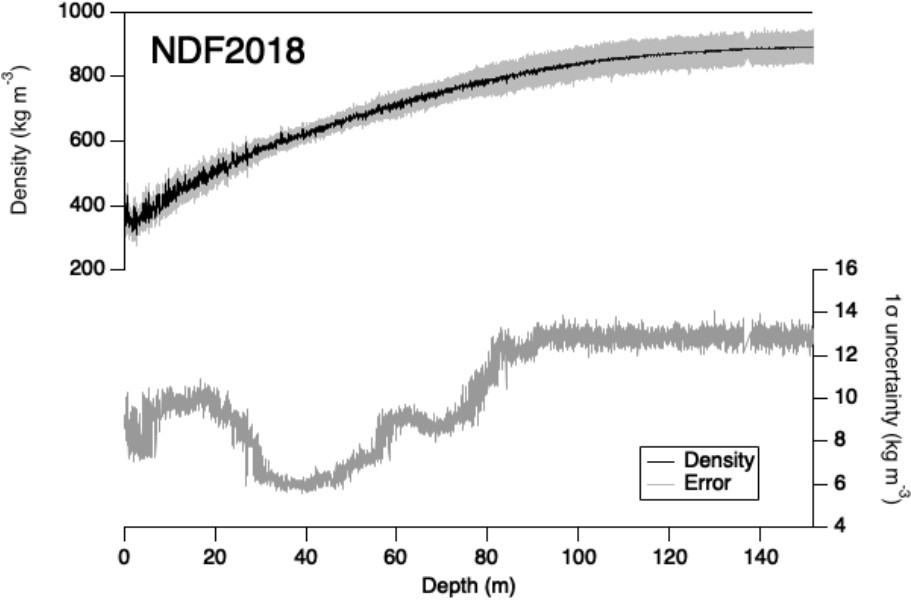


**Figure A2: Example of density uncertainty (NDF2018 core).**

**Data availability**

All data presented in this study is available at the NIPR ADS data repository ([dataset] Oyabu et al. (2022b);
https://ads.nipr.ac.jp/dataset/A20220819-001).

**Author contribution**

SF, IO and KK conceived the idea of this study. HM, KK, SF, FN, MH, TS, KF, YH, IO, KS, HO, NK and ST conducted the
field observations, including ice coring and snow sampling. SF, HM, RI, NK, and MH carried out the laboratory
measurements of ice cores and snow samples. IO and KK developed the analytical method for the SMB estimation, IO
performed the data analyses, and IO, KK, SF, MN, MY, FS and AA interpreted/discussed the data. FP provided Paleochrono
model. IO and KK wrote the manuscript with contributions from all co-authors.

**Financial support**

This study was supported by Japan Society for the Promotion of Science (JSPS) and Ministry of Education, Culture, Sports,
Science and Technology-Japan (MEXT) KAKENHI Grant Numbers 20H04327 to IO, 17H06320 and 20H00639 to KK,



18H05294 to SF, 21221002 and 18H04139 to HM, 25871050 to MH, 18K18176 and 20H04978 to ST, by JST FOREST
     Program (Grant Number JPMJFR216X) to IO and by NIPR Research Projects (Senshin Project and KP305).

**Competing interests**

The authors declare that they have no conflict of interest.

**Acknowledgment**

All field activities were conducted as parts of the Japanese Antarctic Research Expeditions (JARE), managed by the Ministry
     of Education, Culture, Sports, Science and Technology (MEXT) and operated by the National Institute of Polar Research
     (NIPR). We acknowledge all field and laboratory personnel who contributed to obtaining the ice core and snow pit samples,
     field logistics, processing and measurements.

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
