# Peer review of "Temporal variations of surface mass balance over the last 5000 years around Dome Fuji, Dronning Maud Land, East Antarctica"

_Climate of the Past, 2022_

## Author Comment (AC1)

We are grateful to the reviewers for their thorough reviews and valuable comments on our manuscript. Our responses and the planned changes for the revision are explained below. Our replies are in blue and reviewer comments are written in black.

**Comments by Reviewer #1**

Review of Oyabu et al., "Temporal variations of surface mass balance over the last 5000 years around Dome Fuji, Dronning Maud land, East Antarctica."

This study adds a significant dataset to surface mass balance estimations for East Antarctica. As the authors suggest, it is primarily aimed at adding a new stacked dataset to a region with only minimal information currently. It is not an indepth analysis of the mechanisms causing variability in the record/s described. So I have read the manuscript in this spirit assuming further analysis of the dataset is underway. I have primarily minor comments as follows.

The introduction is a good assessment of current knowledge and knowledge gaps.

Line 91 – there is also a first order issue here – that most of the EAIS is difficult to access in a spatially coherent way – so any addition to the dataset such as this is valuable. Thanks, we will add the comment in the revised manuscript.

"There is a primary issue that most of the EAIS is difficult to access in a spatially coherent way. In addition, there are several difficulties in reliable, continuous and long-term SMB reconstructions, particularly from the EAP.  $(1) \cdot \cdot \cdot \cdot$ "

Line 113 – what does 'continentality' mean?

We will replace the word 'continentality' with 'distance from the moisture source (ocean) along the atmospheric pathway'.

Lines 170-175 – could do with a bit of editing for readability.

We will rewrite the paragraph as follows.

**"2.2.2 Relative permittivity at millimeter wave frequencies**

We measured the high-frequency-limit relative permittivity (hereafter relative permittivity, permittivity, or  $\varepsilon$ ) of firn cores using open resonators operating under frequencies from ca. 15 to 40 GHz. The relative permittivities were measured at NIPR and converted to firn densities  $\rho$  (kg m-3) using empirical relations between  $\varepsilon$  and  $\rho$  (Kovacs et al., 1995; Fujita et al., 2014). The detailed method for the measurement of  $\varepsilon$  is described elsewhere (Fujita et al., 2009, 2014, 2016; Saruya et al., 2022). Briefly, a core piece is cut into a slab-shaped sample with a typical thickness of 5–80 mm and a width of 45– 65 mm. The sample is scanned along the depth axis with an open resonator with a 1- $\sigma$  beam diameter of 15–38 mm. This method allows the detection of tensorial values of  $\varepsilon$ , from which the vertical ( $\varepsilon_v$ ) and horizontal ( $\varepsilon_h$ ) components can be simultaneously derived. The measured  $\varepsilon_h$  was converted to  $\rho$  using empirical relations under the temperature of the measurements, either -16±1.5 °C or -30±1.5 °C. The analytical uncertainty of  $\varepsilon_h$  is ±0.005, and the overall uncertainty for the density is estimated to be ~ 6–14 kg m-3 (Appendix A).

For the measurements under -16 °C, we used

 $\rho = -5.556\varepsilon_h^3 - 4.0922\varepsilon_h^2 + 494.14\varepsilon_h - 436.121 \quad (1) \qquad (Fujita et al., 2014).$ This relation was applied for the DF1993, DF1996, DF1999, DFS2010, NDF2013 and S80 cores, which were measured before 2018. Note that we used the published values for the DF1993, DF1999 and DFS2010 cores (Fujita et al., 2016).

For the measurements under -30 °C, we used  $\rho = -20.15\varepsilon_h^3 + 99.801\varepsilon_h^2 + 243.02\varepsilon_h - 220.57 \qquad (2) \qquad (this study).$ This relation was applied for the NDF2018, DFNW, DFSE and NDFN cores, which were measured in 2018–2021."

Section starting at line 259 – there is not enough detail here about these analyses. At what resolution were they measured? Continuous or at interval's and in what cores? What were mean concentrations and were these ok for detection limits?

Ion concentrations were continuously measured on discrete samples at the following resolutions: 5-25 cm for the MD364 core, 2-5 cm (7.70–85.49 m depth) and 20 cm (0–7.70 and 85.49–122.4 m depth) for the DF2001 core, 5-7 cm for the NDF2013 core, 4-6 cm for the S79 core, 5 cm for the S80 core, 4 cm for the Dome Fuji pit, 3 cm for the NDF2018 pit, and 2 cm for the DFS2010, NDF2013, S79 and S80 pits. For ice cores, average concentrations of nssSO42- are 71.8, 105.8, 112.7, 99.6 and 103.4 µg L-1 for the MD364, DF2001, NDF2013, S79 and S80 cores, respectively and the standard deviations of nssSO42- background concentration are 17.4, 28.9, 27.0, 70.7 and 46.1 µg L-1 for the MD364, DF2001, NDF2013, S79 and S80 cores, respectively. We identified volcanic signals if the peak concentrations are larger than mean concentration plus 1 $\sigma$  standard deviation of the background concentration. For snow samples, the mean nssSO42- concentrations are 112.2, 108.7, 112.7, 90.7, and 104.0 µg L-1 for the DFS2010, NDF2013, S79 and S80 pits. The detection limits for the concentrations of SO42- and Na+ are 0.1 and 0.2 µg L-1, respectively, which were sufficient for detecting nssSO42- peaks for the dating.

We will add these descriptions in the revised manuscript.

Section starting at line 275 – this needs an introductory sentence. Explain why you were measuring tritium, and why this is essential to this particular study.

Vast amounts of tritium have been injected into the atmosphere between 1954 and 1963 by nuclear bomb tests, and previous studies showed that snow pits and firn cores in Antarctica have the peak concentration in 1966 (Jouzel et al., 1979; Kamiyama et al., 1989; Fourré et al., 2006). Therefore, tritium data provides a strong age marker for the snow and ice-core samples in this study, for which annual layers cannot be counted.

We add the following sentence at the start of section 2.3.3.

"A peak of tritium content in snow pit and firn core samples in Antarctica provides an age marker at 1966 (Jouzel et al., 1979; Kamiyama et al., 1989; Fourré et al., 2006), which is important for samples from low accumulation areas where annual layer counting is impossible."

- Jouzel, J., Merlivat, L., Pourchet, M., & Lorius, C. (1979). A continuous record of artificial tritium fallout at the South Pole (1954-1978). Earth and Planetary Science Letters, 45(1), 188–200. http://doi.org/10.1016/0012-821X(79)90120-1
- Kamiyama, K., Ageta, Y., & Fujii, Y. (1989). Atmospheric and depositional environments traced from unique chemical compositions of the snow over an inland high plateau, Antarctica. Journal of Geophysical Research D: Atmospheres, 94(D15), 18515–18519. http://doi.org/10.1029/jd094id15p18515
- Fourré, E., Jean-Baptiste, P., Dapoigny, A., Baumier, D., Petit, J. R., & Jouzel, J. (2006). Past and recent tritium levels in Arctic and Antarctic polar caps. Earth and Planetary Science Letters, 245(1-2), 56–64. http://doi.org/10.1016/j.epsl.2006.03.003

Line 283 – depending on resolution...

For low-resolution data, the volcanic peaks in the data may be slightly off from the actual peak position, so we will modify the sentence as follows.

"The depth of a volcanic layer is generally defined at the maximum value of high-resolution data of DEP, ECM or  $nssSO_4^{2-}$  (Hofstede et al., 2004), with depth uncertainty defined by the data resolution."

Line 284 – this is dependent on site resolution/annual accumulation, so this statement is not correct for some sites in Antarctica, e.g. high resolution coastal sites, where the lag can be as low as 6 months. Be more specific.

We cited papers about the lag for relatively recent low-latitude eruptions such as Pinatubo or Tambora in inlands of Greenland (e.g., Crete, Summit) and Antarctica (e.g., South Pole, WAIS Divide). We will clarify the sentence as follows.

"The volcanic signals of low-latitude eruptions are recorded in Antarctic inland snow with a typical lag of 1 to 2 years due to the long-distance transport (Hammer et al., 1980; Cole-Dai and Mosley-Thompson, 1999; Cole-Dai et al., 2000, 2009; Gao et al., 2006)."

Fig. 3. Is it possible to use some colour banding or other visual way to illustrate the common ties in this figure? In shallower cores it is relatively easy to visually pick out the common groups of ties, but in deeper areas this is close to impossible, so the point of the figure is lost. The figure is a promising one, but currently risks losing the readers faith that the figure is designed to clearly allow the reader to discern common ties. It is not possible with just the figure alone (regardless of what is in the supp info). The figure should stand alone in being able to communicate the ties.

We will change the figure to plot the data against age and use colored vertical lines so that the common tie points are vertically aligned and easily compared between the cores. A figure of the same data and colors but plotted against depth is provided in Appendix. We will also correct an error in the original manuscript that three age tie points for the DF2001 core (between 775 and 994 C.E.) were not volcanic but from the matching of ice-core 10Be with tree-

---

## Author Response (AR1)

**Author response to the review of Oyabu et al. "Temporal variations of surface mass balance over the last 5000 years around Dome Fuji, Dronning Maud Land, East Antarctica"**

We are grateful to Dr. Alexey Ekaykin and the reviewers for their thorough reviews on our manuscript. We revised the manuscript following the comments. Our replies are in blue and reviewer comments are written in black.

**Comments by Reviewer #1**

Review of Oyabu et al., "Temporal variations of surface mass balance over the last 5000 years around Dome Fuji, Dronning Maud land, East Antarctica."

This study adds a significant dataset to surface mass balance estimations for East Antarctica. As the authors suggest, it is primarily aimed at adding a new stacked dataset to a region with only minimal information currently. It is not an indepth analysis of the mechanisms causing variability in the record/s described. So I have read the manuscript in this spirit assuming further analysis of the dataset is underway. I have primarily minor comments as follows.

The introduction is a good assessment of current knowledge and knowledge gaps.

Line 91 – there is also a first order issue here – that most of the EAIS is difficult to access in a spatially coherent way – so any addition to the dataset such as this is valuable.
Thanks, we added the comment in the revised manuscript (Line 92 of the track changes version).

*"There is a primary issue that most of the EAIS is difficult to access in a spatially coherent way. In addition, there are several difficulties in reliable, continuous and long-term SMB reconstructions, particularly from the EAP. (1) · · · · ·"*

Line 113 – what does 'continentality' mean?
We replaced the word 'continentality' with 'distance from the moisture source (ocean) along the atmospheric pathway' (Line 115 – 116 of the track changes version).

Lines 170-175 – could do with a bit of editing for readability.
We rewrote the paragraph as follows.

*"2.2.2 Relative permittivity at millimeter wave frequencies*

*We measured the high-frequency-limit relative permittivity (hereafter relative permittivity, permittivity, or ε) of firn cores using open resonators operating under frequencies from ca. 15 to 40 GHz. The relative permittivities were measured at NIPR and converted to firn densities ρ (kg m⁻³) using empirical relations between ε and ρ (Kovacs et al., 1995; Fujita et al., 2014). The detailed method for the measurement of ε is described elsewhere (Fujita et al., 2009, 2014, 2016; Saruya et al., 2022). Briefly, a core piece is cut into a slab-shaped sample with a typical thickness of 5–80 mm and a width of 45–*

*65 mm. The sample is scanned along the depth axis with an open resonator with a 1-σ beam diameter of 15–38 mm. This method allows the detection of tensorial values of ε, from which the vertical ($\varepsilon_v$) and horizontal ($\varepsilon_h$) components can be simultaneously derived. The measured $\varepsilon_h$ was converted to ρ using empirical relations under the temperature of the measurements, either -16±1.5 °C or -30±1.5 °C. The analytical uncertainty of $\varepsilon_h$ is ±0.005, and the overall uncertainty for the density is estimated to be ~ 6–14 kg m⁻³ (Appendix A).*

*For the measurements under -16 °C, we used*

$$\rho = -5.556\varepsilon_h^3 - 4.0922\varepsilon_h^2 + 494.14\varepsilon_h - 436.121 \qquad (1) \qquad (Fujita\ et\ al.,\ 2014).$$

*This relation was applied for the DF1993, DF1996, DF1999, DFS2010, NDF2013 and S80 cores, which were measured before 2018. Note that we used the published values for the DF1993, DF1999 and DFS2010 cores (Fujita et al., 2016).*

*For the measurements under -30 °C,   we used*

$$\rho = -20.15\varepsilon_h^3 + 99.801\varepsilon_h^2 + 243.02\varepsilon_h - 220.57 \qquad (2) \qquad (this\ study).$$

*This relation was applied for the NDF2018, DFNW, DFSE and NDFN cores, which were measured in 2018–2021."*

Section starting at line 259 – there is not enough detail here about these analyses. At what resolution were they measured? Continuous or at interval's and in what cores? What were mean concentrations and were these ok for detection limits?

Ion concentrations were continuously measured on discrete samples at the following resolutions: 5–25 cm for the MD364 core, 2–5 cm (7.70–85.49 m depth) and 20 cm (0–7.70 and 85.49–122.4 m depth) for the DF2001 core, 5–7 cm for the NDF2013 core, 4–6 cm for the S79 core, 5 cm for the S80 core, 4 cm for the Dome Fuji pit, 3 cm for the NDF2018 pit, and 2 cm for the DFS2010, NDF2013, S79 and S80 pits. For ice cores, average concentrations of $nssSO_4^{2-}$ are 71.8, 105.8, 112.7, 99.6 and 103.4 μg L⁻¹ for the MD364, DF2001, NDF2013, S79 and S80 cores, respectively and the standard deviations of $nssSO_4^{2-}$ background concentration are 17.4, 28.9, 27.0, 70.7 and 46.1 μg L⁻¹ for the MD364, DF2001, NDF2013, S79 and S80 cores, respectively. We identified volcanic signals if the peak concentrations are larger than mean concentration plus 1σ standard deviation of the background concentration. For snow samples, the mean $nssSO_4^{2-}$ concentrations are 112.2, 108.7, 112.7, 90.7, and 104.0 μg L⁻¹ for the DFS2010, NDF2013, S79 and S80 pits. The detection limits for the concentrations of $SO_4^{2-}$ and $Na^+$ are 0.1 and 0.2 μg L⁻¹, respectively, which were sufficient for detecting $nssSO_4^{2-}$ peaks for the dating.

We added these descriptions in the revised manuscript (section 2.3.2).

Section starting at line 275 – this needs an introductory sentence. Explain why you were measuring tritium, and why this is essential to this particular study.

Vast amounts of tritium have been injected into the atmosphere between 1954 and 1963 by nuclear bomb tests, and previous studies showed that snow pits and firn cores in Antarctica have the peak concentration in 1966 (Jouzel et al., 1979; Kamiyama et al., 1989; Fourré et al., 2006). Therefore, tritium data provides a strong age marker for the snow and ice-core samples in this study, for which annual layers cannot be counted.

We added the following sentence at the start of section 2.3.3.

*"A peak of tritium content in snow pit and firn core samples in Antarctica provides an age marker at 1966 (Jouzel et al., 1979; Kamiyama et al., 1989; Fourré et al., 2006), which is important for samples from low accumulation areas where annual layer counting is impossible."*

Jouzel, J., Merlivat, L., Pourchet, M., & Lorius, C. (1979). A continuous record of artificial tritium fallout at the South Pole (1954-1978). Earth and Planetary Science Letters, 45(1), 188–200. http://doi.org/10.1016/0012-821X(79)90120-1

Kamiyama, K., Ageta, Y., & Fujii, Y. (1989). Atmospheric and depositional environments traced from unique chemical compositions of the snow over an inland high plateau, Antarctica. Journal of Geophysical Research D: Atmospheres, 94(D15), 18515–18519. http://doi.org/10.1029/jd094id15p18515

Fourré, E., Jean-Baptiste, P., Dapoigny, A., Baumier, D., Petit, J. R., & Jouzel, J. (2006). Past and recent tritium levels in Arctic and Antarctic polar caps. Earth and Planetary Science Letters, 245(1-2), 56–64. http://doi.org/10.1016/j.epsl.2006.03.003

Line 283 – depending on resolution…

For low-resolution data, the volcanic peaks in the data may be slightly off from the actual peak position, so we modified the sentence as follows (Line 318 of the track changes version).

*"The depth of a volcanic layer is generally defined at the maximum value of high-resolution data of DEP, ECM or nssSO$_4^{2-}$ (Hofstede et al., 2004), with depth uncertainty defined by the data resolution."*

Line 284 – this is dependent on site resolution/annual accumulation, so this statement is not correct for some sites in Antarctica, e.g. high resolution coastal sites, where the lag can be as low as 6 months. Be more specific.

We cited papers about the lag for relatively recent low-latitude eruptions such as Pinatubo or Tambora in inlands of Greenland (e.g., Crete, Summit) and Antarctica (e.g., South Pole, WAIS Divide). We clarified the sentence as follows (Line 319 of the track changes version).

*"The volcanic signals of low-latitude eruptions are recorded in Antarctic inland snow with a typical lag of 1 to 2 years due to the long-distance transport (Hammer et al., 1980; Cole-Dai and Mosley-Thompson, 1999; Cole-Dai et al., 2000, 2009; Gao et al., 2006)."*

Fig. 3. Is it possible to use some colour banding or other visual way to illustrate the common ties in this figure? In shallower cores it is relatively easy to visually pick out the common groups of ties, but in deeper areas this is close to impossible, so the point of the figure is lost. The figure is a promising one, but currently risks losing the readers faith that the figure is designed to clearly allow the reader to discern common ties. It is not possible with just the figure alone (regardless of what is in the supp info). The figure should stand alone in being able to communicate the ties.

We changed the figure to plot the data against age and use colored vertical lines so that the common tie points are vertically aligned and easily compared between the cores. A figure of the same data and colors but plotted against

depth is provided in Appendix. We also corrected an error in the original manuscript that three age tie points for the DF2001 core (between 775 and 994 C.E.) were not volcanic but from the matching of ice-core [10]Be with tree-ring [14]C as published by Oyabu et al. (2022). We added the explanation in the text (Line 348 of the track changes version).

[Figure]

Figure 3: DEP, ECM or nssSO$_4^{2-}$ for our ice cores and sulfur concentration of the WAIS Divide core (Cole-Dai, 2014a, b). Previously published data are nssSO$_4^{2-}$ for 1900 – 0 C.E. for the DF2001 core (right axis) (Motizuki et al., 2014), ECM for 1900 C.E. – 500 B.C.E. for the DF1993 core (left axis), ECM for 900 – 3200 B.C.E. for the DF1 core (left axis) (Fujita et al., 2015), and DEP for 1200 – 3200 B.C.E. for the DF2 core (right axis) (Fujita et al., 2015). Other data are previously unpublished (see text). Vertical colored lines indicate volcanic tie points. Three black dashed lines for the DF2001 core indicate additional age tie points from published [10]Be-[14]C matching (Oyabu et al., 2022). Age scales between the tie points were calculated with cubic spline interpolation. See Fig. B1 for the same data plotted against depth.

[Figure]

Figure B1: Same data and colored lines as Figure 3, but plotted against depth.

Fig. 4. Other double Pinatubo peaks have been detected elsewhere in east antarctica. E.g. see Plummer et al., 2012 or Crockart et al., 2021 (both in Climate of the Past). There is also always a chance that this relatively small eruption is confused with another unidentified regional eruption.

Thank you for the other examples. We modified the sentence at line 366 of the track changes version, as follows.

"*We do not use the Pinatubo eruption for the age tie point in the Dome Fuji pit data, because* there are three possible nssSO$_4^{2-}$ peaks for the eruption within ~0.5 m interval (around 1.0 m), *making the precise depth-age assignment impossible. Unidentified regional eruptions may have left impurities peaks near the Pinatubo depth, as have been found elsewhere in East Antarctica (Cole-Dai et al., 1997; Plummer et al., 2012; Crockart et al., 2021)."*

Line 362 – any age errors?

The 5152 years is constrained by a volcanic tie point, and its age error is ±10 years, mostly coming from the uncertainty of the WAIS Divide chronology (WD2014). We added "±10" in the revised manuscript (Line 411 of the track changes version).

Line 383 – rephrase, this is confusing prose. E.g. "longer to the north and smaller to the south"

We rephrased the sentence as suggested (Line 433 of the track changes version).

Line 387 – or larger variability in orographic deposition

We agreed and changed the sentence to *"The larger variability in the depositional environment may be expected from the horizontal ice flow (4.1 m yr$^{-1}$) over rough bedrock topography, creating temporal variations in local surface topography (e.g., surface slope and convex/concave curvatures) (Fujita et al., 2002; Kahle et al., 2021), or from the variable wind direction for deposition in combination with surface topography (Urbini et al., 2008)."* (Line 436 of the track changes version)

Section starting at line 398 – you need to explain why you chose these periods. Presumably for dating/volcanic tie reasons, but to do a good analysis of climatologically why there are differences, you also need good climatological reasons for the separation of your different time periods.

We chose the three time periods for the following climatological reasons:

(1) 1461–1816 C.E., because it is the most accurately constrained period by the volcanic years common for all the studied ice cores, and also a good representation of the preindustrial late Holocene for averaging SMB without decadal noise.

(2) 0–1850 C.E., because the 0 C.E. is the start year of the "PAGES 2k network" datasets, which is connected to the Paleoclimate Modelling Intercomparison Project (PMIP), for providing the baseline information about natural climate variability (e.g., Martrat et al., 2019). The 1850 C.E. is chosen for investigating the average preindustrial SMB within the PAGES 2k framework.

(3) 2000 B.C.E–0 C.E., to compare the averages with those in the recent 2k period for the individual sites, and also to examine the consistency of the spatial gradient of SMB over the multi-millennial timescales.

We added the above information to the revised manuscript (around line 455 of the track changes version).

Martrat, B., Eggleston, S., Abram, N., Bothe, O., Linderholm, H., McGregor, H., Neukom, R., Phipps, S., and St George, S.: The PAGES 2k Network: Understanding the climate of the Common Era (past 2000 years), Geophysical Research Abstracts, Vol. 21, EGU2019-16976, 2019, EGU General Assembly 2019.

Figure 7 and Table 3. As best I can tell, these two elements display the same information. Remove one or the other, probably the table to supp info.

We removed Table 3 and include its information in the supplementary datasets.

Section around line 505. This has also been reported at Dome C, with the shifting Dome position related to small variations in wind direction, which have further been related to the frequency and position of mid-latitude atmospheric blocking in the southern ocean. See papers by Frezzotti and Scarchilli, and also Masson, Pook et al. Also recent work by Jonathan Wille, John Turner and Danielle Udy on the influence of meridional mid-latitude atmospheric variability and events on east Antarctic SMB and ice cores.

Thank you for the information. In our manuscript, we discussed that the possible movement of the dome position might cause slight differences in the long-term SMB trends of the three sites on the glacial-interglacial time scale. Thus, our discussion does not exactly match that of Urbini et al. (2008), that the spatio-temporal changes in SMB might influence the dome position on decadal to centennial time scales. However, the discussion for Dome C and Talos Dome would be useful for the readers, thus we included the following part in the revised manuscript (Line 565 of the track changes version).

*"The difference in the trends might be related to possible long-term migration of the dome summit position in response to glacial-interglacial changes in SMB and grounding line position (Saito, 2002; Parrenin et al., 2016). The dome migration could also possibly be a consequence of spatial and temporal variations of SMB, which may be influenced by meridional mid-latitude atmospheric variability and precipitation events (Massom et al., 2004; Scarchilli et al., 2011, Tuner et al., 2019), as has been discussed for Dome C and Talos Dome (Urbini et al., 2008)."*

Urbini, S., Frezzotti, M., Gandolfi, S., Vincent, C., Scarchilli, C., Vittuari, L., and Fily, M.: Historical behaviour of Dome C and Talos Dome (East Antarctica) as investigated by snow accumulation and ice velocity measurements. Glob. Planet. Change, 60(3-4), 576–588, http://doi.org/10.1016/j.gloplacha.2007.08.002, 2008.

Scarchilli, C., Frezzotti, M., & Ruti, P. M.: Snow precipitation at four ice core sites in East Antarctica: Provenance, seasonality and blocking factors. Climate Dynamics, 37(9-10), 2107–2125. http://doi.org/10.1007/s00382-010-0946-4, 2011.

Massom, R. A., Pook, M. J., Comiso, J. C., Adams, N., Turner, J., Lachlan-Cope, T., & Gibson, T. T.: Precipitation over the Interior East Antarctic Ice Sheet Related to Midlatitude Blocking-High Activity. Journal of Climate, 17(10), 1914–1928. http://doi.org/10.1175/1520-0442(2004)017<1914:POTIEA>2.0.CO;2, 2004.

Turner, J., Phillips, T., Thamban, M., Rahaman, W., Marshall, G. J., Wille, J. D., Favier, V., Winton, V. H. L., Thomas, E., Wang, Z., van den Broeke, M., Hosking, J. S., Lachlan-Cope, T.: The dominant role of extreme precipitation events in Antarctic snowfall variability. Geophysical Research Letters, 46,3502–3511. https://doi.org/10.1029/2018GL081517, 2019.

The discussion is well written for what is predominantly data focused paper. I think a bit more discussion is required in the section at lines 675-680 around why GHG and ozone depletion might increase moisture content over the Southern Ocean. Readers may be aware of the theories, but they should still be elaborated on here, as this increase in SMB is one of the primary findings surely? Also – I'm not sure whether GHG and Ozone depeletion can be easily related to interior east Antarctica yet. Are there any papers you can cite specifically about interior SMB changes and GHG/ozone?

It seems straightforward in the global climate models that GHG increase warms the surface temperature and thus the atmospheric moisture content over the whole Antarctic continent, leading to the precipitation increase (e.g., Previdi

and Polvani, 2016, 2017). The link between ozone depletion and Antarctic SMB increase is more complex and involves atmospheric dynamics. Recent studies suggest two mechanisms to enhance the meridional atmospheric transport (moisture fluxes) toward Antarctica (Lenaerts et al., 2018; Chemke et al., 2020). The first is by strengthening the zonal wind in the lower troposphere, which is balanced by the increase of meridional flow. The second is by enhancing the barotropic instability, which increases the poleward eddy moisture fluxes.

There have been no studies to particularly focus on the Antarctic interior SMB and the anthropogenic forcings, but we can see in the cited papers that both GHG and ozone effects seem to create circumpolar patterns of SMB increase over the Antarctic continent, with decreasing magnitude of change towards interior (Chemke et al., 2020; Dunmire et al., 2022). We also note that Chemke et al. (2020) separated for the first time the effects of stratospheric ozone depletion, an increase of ozone-depleting substances (ODSs), and tropospheric ozone change, and suggested that the ODSs and tropospheric ozone effects are not longitudinally uniform.

For the revised manuscript, we added the following (Line 744 of the track changes version).
*"As previous studies have suggested, the significant increase in accumulation rate in the 20th century may be attributed to anthropogenic forcings such as increased atmospheric GHG concentrations and stratospheric ozone depletion (e.g., Medley and Thomas, 2019). Climate models suggest that enhanced GHG radiative forcing would increase temperature and, thus, the moisture content in the atmosphere over the Southern Ocean and Antarctica, leading to the increase of Antarctic precipitation (e.g., Previdi and Polvani, 2016). The ozone depletion is suggested to strengthen the mean zonal wind at mid to high latitudes in the lower troposphere, which is balanced by a stronger mean meridional wind (Chemke et al., 2020). In addition, ozone depletion would enhance barotropic instability, which also increases the poleward eddy moisture fluxes (Chemke et al., 2020)."*

Previdi, M., and Polvani, L. M.: Anthropogenic impact on Antarctic surface mass balance, currently masked by natural variability, to emerge by mid-century, Environ. Res. Lett., 11(9), 094001. http://doi.org/10.1088/1748-9326/11/9/094001, 2016.

Previdi, M., and Polvani, L. M.: Impact of the Montreal Protocol on Antarctic surface mass balance and implications for global sea level rise, J. Clim., 30(18), 7247–7253. http://doi.org/10.1175/JCLI-D-17-0027.1, 2017.

Lenaerts, J. T. M., Fyke, J., and Medley, B.: The Signature of Ozone Depletion in Recent Antarctic Precipitation Change: A Study With the Community Earth System Model, Geophys. Res. Lett., 45, 12-931-912-939, https://doi.org/10.1029/2018GL078608, 2018.

Chemke, R., Previdi, M., England, M. R., and Polvani, L. M.: Distinguishing the impacts of ozone and ozone-depleting substances on the recent increase in Antarctic surface mass balance, The Cryosphere, 14, 4135-4144, https://doi.org/10.5194/tc-14-4135-2020, 2020.

Dunmire, D., Lenaerts, J. T. M., Datta, R. T., and Gorte, T.: Antarctic surface climate and surface mass balance in the Community Earth System Model version 2 during the satellite era and into the future (1979–2100), The Cryosphere, 16, 4163–4184, https://doi.org/10.5194/tc-16-4163-2022, 202

**Comments by Reviewer #2**

This work represents a new time-series of snow mass balance for the last 5000 years in the vicinity of Dome Fuji (central East Antarctica). Since in this region there is a huge lack of such data, this work is a very important contribution to the understanding of the factors controlling the behavior of SMB in East Antarctica. The authors present in details the processes of obtaining the SMB data including the involved uncertainties. They also compare the newly obtained timeseries with the SMB data from the other Antarctic regions, as well as with other climatic records of the Southern Hemisphere.

General comment: in your manuscript you discuss the local air temperature as an important factor governing the SMB (e.g., section 4.1). In view of this, it would be useful to present in this paper the stable water isotopes records from the same cores, as temperature proxies. This would also allow to calculate the isotope-SMB sensitivity, which would be relevant to the other studies. Please consider this possibility.

There are 6 cores with stable water isotope data (MD364, DF1, DF2001, NDF2013, S79 and S80). We examined the correlation between the SMB and $\delta^{18}O$ after averaging the $\delta^{18}O$ data over the same time intervals of the SMB reconstruction for each core. We found significant correlations from relatively short cores from NDF2013 and S79, but the length of the NDF2013, S79 and S80 look too short to examine the relationship between the SMB and $\delta^{18}O$. The other cores do not show significant correlations.

We can expect that the relationships between the SMB and $\delta^{18}O$ in the DF1 and DF2001 are similar to each other because of their close proximity (both at the Dome Fuji station with a distance of only 43 m). However, the relationship between SMB and $\delta^{18}O$ looks rather different for the two cores. For example, while the $\delta^{18}O$ of both cores show an increasing trend over the last 200 years, only the DF2001 core shows an increasing trend in the reconstructed SMB. Possible reasons for the dissimilarity may be the low resolution of the SMB reconstruction as well as the difficulty in the precise SMB reconstructions for the shallowest part (due to the fragility of the samples). In addition, there is no significant correlation between SMB and $\delta^{18}O$ in both cores. We think it is difficult at this stage to show robust results, and further analyses would be necessary, which is beyond the scope of this study. Thus, we refrain from adding the stable water isotope records and discussion to the manuscript. We show the figures below for review purpose.

[Figure]

Figure: (left) Time series of the SMB (black dotted line) and δ¹⁸O (blue) records of the MD364, DF1, DF2001, NDF2013, S79 and S80 cores. Solid lines with pale blue indicate all δ¹⁸O data and the lines with blue indicate the δ¹⁸O data averaged over the same intervals of the SMB reconstruction. (right) Scatter plot of δ¹⁸O vs. SMB.

Minor comments:

Table 1: I am not sure if the last column is really necessary. The number of JARE campaign really tells nothing to a reader. The observation date is enough.

Some international readers would find the information useful to quickly recognize which cores are from which campaign (also, the year/date is sometimes confusing for Austral summer to recognize the same sampling campaign), so we would like to keep them in the table.

Lines 158-160: it should be possible to evaluate the error of the bulk density by comparing the density values measured in the same depth intervals in different (but closely located, e.g., in the vicinity of the DF station) cores. If we assume that the density-depth profiles are constant in time (Sorge's law), then the density at the same depths should be the same in different neighboring cores, and the difference between them would be explained by the measurement errors and the spatial variability.

Thank you for your suggestion. There are two bulk density data from the DF1993 and DF2001 cores drilled at Dome Fuji station. The distance between the two boreholes is 43 m at the surface. We resampled the density data at 0.5 m intervals from 10 to 112 m where both data are available (191 intervals), and evaluated the variability of densities by calculating the pooled standard deviation ($S_p$) as the square root of the summed squared deviations of the DF1993 and DF2001 density data from the respective means for the 0.5-m intervals, divided by the number of depth intervals:

$$S_p = \sqrt{\frac{\sum_{i=1}^{382}(\rho - \overline{\rho})^2}{191}},$$

where $\rho$ is density for the 382 individual data, and $\overline{\rho}$ is the mean of DF1993 and DF2001 for the 191 depth intervals.

We found that $S_p$ is 16 kg m$^{-3}$, which is comparable to the independently estimated error of 15 kg m$^{-3}$. Thus, we kept the original uncertainty estimate for the bulk density and added the above description as a validation at the DF site (around line 160 of the track changes version).

Line 217: d is depth in m.

We corrected this in the revised manuscript (Line 238 of the track changes version).

Line 654: to my knowledge, the 1458 eruption that was previously interpreted as Kuwae, now is rather interpreted as unknown event (Hartman, L.H., Kurbatov, A.V., Winski, D.A., Cruz-Uribe, A.M., Davies, S.M., Dunbar, N.W., Iverson, N.A., Aydin, M., Fegyveresi, J.M., Ferris, D.G., Fudge, T.J., Osterberg, E.C., Hargreaves, G.M. and Yates, M.G. (2019). Volcanic glass properties from 1459 C.E. volcanic event in South Pole ice core dismiss Kuwae caldera as a potential source. Nature Scientific Reports 9(14437), 1-7. doi: 10.1038/s41598-019-50939-x).

Thank you for the information. We replaced Kuwae with unknown in the revised manuscript with the reference (Line 720 of the track changes version and Figure 13). The volcanic forcing of Sigl et al. (2022) (used in this study) may be weaker for this eruption, so maybe the negative SMB anomaly in this period should not be caused by the volcanic forcing.